# *Staphylococcus epidermidis* uses the SrrAB regulatory system to modulate oxidative stress and intracellular survival in mouse macrophage cell line Ana-1

Chunjing Zhao,[1] Zongkai Bai,[1] Xiaoting Chen,[1] Shuangjie Shang,[1] Baitong Shen,[1] Li Bai,[1] Di Qu,[2] Yang Wu,[2] Youcong Wu[1]

**ABSTRACT**   The two-component system (TCS) SrrAB responds to oxidative stress in *Staphylococcus epidermidis*. A *srrAB* deletion mutant (Δ*srrAB*) was constructed using *S. epidermidis* strain 1457 (SE1457) as the parent strain to study its regulatory function in oxidative stress. Compared to SE1457, the viable cell counts of the Δ*srrAB* mutant significantly decreased in the post-stationary phase culture, coinciding with a sharp increase in reactive oxidative species (ROS) accumulation. The impaired growth of the Δ*srrAB* mutant was partially restored by shifting the culture from oxic to microaerobic conditions. Consistently, growth of the Δ*srrAB* mutant in tryptone soy broth (TSB) medium containing $H_2O_2$ was notably inhibited compared to parent strain SE1457, and the mutant showed significantly decreased resistance (100- to 1,000-fold) to $H_2O_2$ and cumene hydroperoxide in both oxic and microaerobic conditions, which was fully rescued by the addition of ROS inhibitor 2,2-dipyridyl. Furthermore, the deletion of *srrAB* resulted in decreased intracellular survival in the Ana-1 macrophages, likely due to intracellular ROS accumulation. The complementation of *srrAB* in the Δ*srrAB* mutant restored ROS resistance and intracellular survival to wild-type levels. RNA-seq analysis revealed that *srrAB* deletion affected the transcription levels of 610 genes, including those involved in oxidative stress, respiratory and energy metabolism, and transition ion homeostasis. These findings were corroborated by quantitative real-time reverse transcription-PCR. In the Δ*srrAB* mutant, expressions of ROS-scavenging genes *katA*, *ahpC*, *scdA*, *serp1797*, and *serp0483* were downregulated compared to SE1457. Electrophoretic mobility shift assay further demonstrated phosphorylated SrrA bound to the promoter regions of *srrAB*, *katA*, *ahpC*, *scdA*, *serp1797*, and *serp0483* genes. This study elucidates that in *S. epidermidis*, SrrAB is the critical TCS to sense and respond to the oxidants, directly regulating transcription levels of the genes involved in ROS scavenging and ion homeostasis, thereby facilitating *S. epidermidis* detoxification of ROS and adaptation to the commensal environment.

**IMPORTANCE**   *Staphylococcus epidermidis* in the human skin and mucous microbiome is a leading cause of hospital-acquired infection, whereas the mechanism by which it inhabits, adapts, and further results in infection is not well known. In this study, we found that the two-component regulatory system SrrAB directly regulates transcription levels of the genes involved in reactive oxidative species (ROS) scavenging and ion homeostasis in *S. epidermidis*, influencing ROS accumulation during growth, thereby facilitating detoxification of ROS and adaptation to the commensal environment. This work provides new molecular insight into the mechanisms of SrrAB in regulating resistance and intracellular viability against oxidative stress in *S. epidermidis*.

**KEYWORDS**   Staphylococcal respiratory response, *Staphylococcus epidermidis*, oxidative stress, macrophage, ROS

Address correspondence to Youcong Wu, wuyoucong@dali.edu.cn, or Yang Wu, yangwu@fudan.edu.cn.

The authors declare no conflict of interest.

$S$ taphylococcus epidermidis, a coagulase-negative bacterium, is a constituent of the human skin and mucosa microbiota (1). It is also one of the most common causes of implant-associated infections. It is a canonical opportunistic biofilm former, particularly in individuals with indwelling medical devices (such as prostheses, catheters, or heart valves) and those with compromised immune systems (2, 3). Staphylococci, being facultative anaerobes, can obtain energy for growth via either respiratory or fermentative pathways. However, oxygen concentrations vary significantly between healthy (ranging from 19.7% to 1.5%; normoxia) and infected tissues (below 1%) (4, 5). Meanwhile, reactive oxidative species (ROS), including hydrogen peroxide ($H_2O_2$), superoxide anion radicals ($O_2^-$), and hydroxyl radicals ($OH^-$), are generated during the aerobic cellular metabolism or phagocytosis of microbes (6). In response to this challenge, staphylococci have evolved various protective, detoxifying, and repair mechanisms regulated by a network of regulators such as two-component regulatory systems (TCSs) (7), PerR (8), SarA (9), and MgrA (10), among others. Adaptation to environmental changes and the ability to cope with oxidant stress from phagocytes are pivotal for bacterial pathogenesis (11).

How S. epidermidis senses and responds to oxidative stress remains largely uncertain. AbfR is the first described sensor of oxidative stress in S. epidermidis and regulates oxidative stress responses, bacterial aggregation, and biofilm formation (12). Unlike S. epidermidis, Staphylococcus aureus employs a diverse array of antioxidant tools, both enzymatic and non-enzymatic, to combat oxidative stress or facilitate the detoxification of ROS (13, 14). The carotenoid pigment, for instance, serves as a potent antioxidant shielding S. aureus against ROS. One study (15) has demonstrated that non-pigmented S. aureus mutants exhibit diminished virulence and survival compared to pigmented wild-type strains in mouse infection models. In addition to pigments, most staphylococci possess several detoxifying enzymes that scavenge ROS. The heme-containing tetrameric catalase encoded by the katA gene decomposes $H_2O_2$ into water and oxygen (16). The alkyl hydroperoxide reductase encoded by the ahpC gene catalyzes the reduction of organic hydroperoxides or $H_2O_2$ to alcohols and/or water in an NADH-dependent manner (17). While KatA primarily confers resistance toward $H_2O_2$, AphC extends resistance to a broader spectrum of ROS in S. aureus (18). The katA and ahpC genes are negatively regulated by PerR due to the putative PerR box presence in both genes' promoter regions (19). ScdA, an iron-sulfur cluster protein, repairs Fe-S clusters, aiding in the recovery of aconitase and fumarase activity post-oxidative damage; inactivation of scdA renders S. aureus more susceptible to $H_2O_2$ (20). Superoxide dismutases (SODs) are metalloenzymes that catalyze the dismutation of $O_2^-$ to oxygen and $H_2O_2$, which can be further reduced to water and oxygen by KatA and AhpC enzymes (21). S. aureus possesses two monocistronic SOD genes, sodA and sodM, while S. epidermidis lacks sodM. Both SODs play crucial roles in maintaining cell viability under ROS stress (22). DNA-binding proteins from starved cells (Dps) belong to the ferritin superfamily, which inhibits Fenton Chemistry and reduces $OH^-$ formation (23). Additionally, transition metal ions (such as $Fe^{2+}$, $Cu^{2+}$, $Mn^{2+}$, and $Zn^{2+}$) serve as enzyme cofactors and are essential for electron transfer; thus, metal ion transportations are tightly modulated to maintain their appropriate intracellular concentrations and to facilitate ROS generation via Fenton chemistry (24).

The TCS SrrAB serves as a "sentinel," sensing environmental oxidants and transducing these signals to regulators that modulate the transcription of defense genes in response to the challenge (25–28). Mashruwala et al. (26) reported that in S. aureus (USA300-LAC strain), SrrAB positively influences the transcription of genes encoding $H_2O_2$ resistance factors (katA, ahpC, dps, and scdA) during periods of high dioxygen-dependent respiratory activity, while exerting a negative influence in the absence of respiration. Moreover, purified SrrA was found to bind specifically to the dps promoter region, suggesting a direct role in dps transcription. However, Oogai et al. (28) observed that SrrAB negatively regulates katA and dps expression in the presence or absence of $H_2O_2$, and a srrA-inactivated mutant displayed enhanced $H_2O_2$ resistance compared

to the wild-type strain (*S. aureus* MW2). These findings highlight the diversity in the proposed models of SrrAB regulon's role in regulating oxidative stress, even among the different species and strains of the same *S. aureus*.

Bioinformatics analysis revealed that although the SrrA/SrrB proteins in SE1457 share approximately 90%/70% identity with that in *S. aureus* strain Mu3 at the amino acid level, there are variations in the extracellular sensor domain of SrrB proteins (I34V, A35S, M40I, R53K, R57K, E81D, M89I, I91M, A100S, V113I, V131I, T143S, I145L, and S157A) and in the CheY homologous receiver domain of SrrA proteins (M20L, M43L, T66S, and S122T) in the two species, indicating that there may be differences between *S. epidermidis* and *S. aureus* SrrAB functions in regulating biological phenotypes and pathogenesis. A previous study by our group has shown that SrrAB in *S. epidermidis* responds to oxygen stress and modulates biofilm formation in an *ica*-dependent manner (27). Furthermore, *S. epidermidis* SrrAB regulates bacterial growth in response to environmental oxygen concentration: it positively regulates *qoxBACD* transcription under oxic conditions while modulating fermentation processes and DNA replication via the *pflBA* operon and *nrdDG* under microaerobic conditions. However, the mechanisms by which *S. epidermidis* SrrAB senses and responds to oxidative stress remain largely elusive. In this study, we aim to elucidate the role of SrrAB in regulating resistance and intracellular viability against oxidative stress in *S. epidermidis,* shedding light on novel aspects of its function.

## RESULTS

### *S. epidermidis* SrrAB responded to oxidative stress

To assess whether SrrAB responds to oxidative stress, researchers analyzed the transcription of *srrAB* in the *S. epidermidis* strain 1457 (SE1457) exposed to different concentrations of $H_2O_2$ and cumene hydroperoxide (CHP) using quantitative real-time reverse transcription-PCR (qRT-PCR). Following a 30 min treatment with 150 µM $H_2O_2$, both *srrA* and *srrB* expression displayed approximately eightfold and sevenfold increases, respectively. Similarly, treatment with 150 µM CHP resulted in a 57-fold increase in *srrA* expression and a 17-fold increase in *srrB* expression (Fig. 1). These findings suggest that *S. epidermidis* SrrAB displays functions in bacterial adaptation to oxidative stress.

### Deletion of *srrAB* decreased viable cells in the culture

To assess the oxidative stress response of *srrAB* in *S. epidermidis*, a *srrAB* deletion mutant of the SE1457 strain was constructed by allelic replacement using the temperature-sensitive plasmid pKOR1 (Fig. S1). The resulting *srrAB* mutant, designated Δ*srrAB*, was verified by PCR, qRT-PCR, and sequencing.

We observed that overnight cultures of the Δ*srrAB* mutant became more transparent than those of the parent strain SE1457 and complementation strain Δ*srrAB*(pCN51-*srrAB*) when kept at room temperature for several days (Fig. S2A). Furthermore, bacterial viability assay revealed a significant decrease (a maximum of $10^4$ folds) in viable cells in the cultures of the Δ*srrAB* mutant compared to SE1457 and Δ*srrAB*(pCN51-*srrAB*), particularly after 24 h (Fig. S2).

### Shifting to microaerobic conditions alleviated growth defects of the Δ*srrAB* mutant

The density of the bacterial population is different, and the resistance against the oxidants is different. To compare the subtle effects of different initial inoculum concentrations of the *srrAB* deletion mutant on oxidative stress, researchers seeded inoculum concentrations ranging from $10^2$ to $10^6$ CFU/mL. When the initial inoculum was lower, the growth of the *srrAB* deletion mutant was more delayed in the lag phase, and the turbidity ($OD_{600}$) in the decline phase (18–20 h time points) decreased significantly faster compared to that of SE1457 under the same conditions. Under oxic conditions ($+O_2$), the growth of the Δ*srrAB* mutant was delayed by about 3–4 h, entering into the log phase and stationary phase compared to the parent strain SE1457. However, under

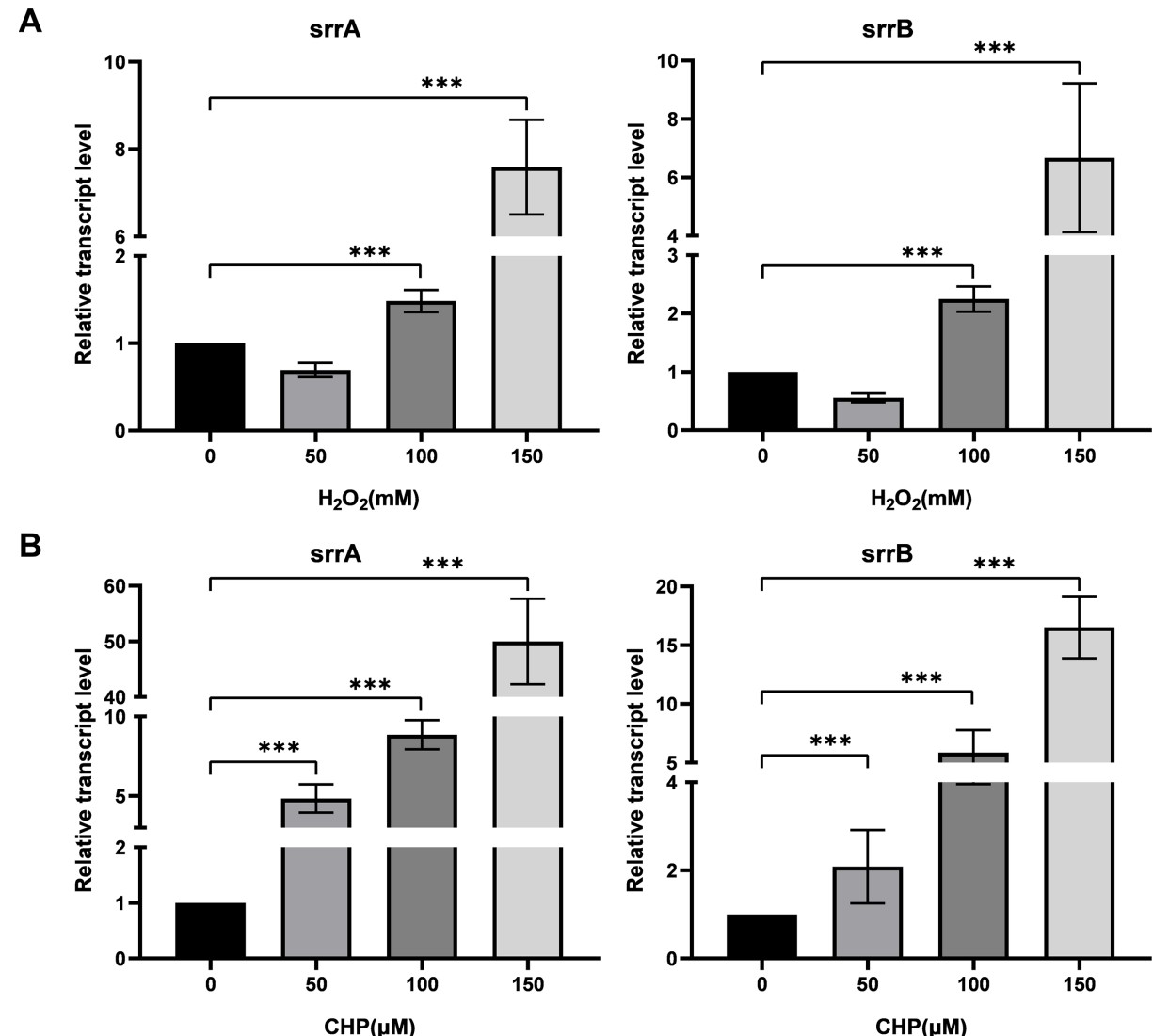

**FIG 1** Transcriptional levels of *srrA* and *srrB* in SE1457 under oxidative stress. After culturing for 4 h, *S. epidermidis* strain 1457 was treated with different concentrations of $H_2O_2$ (A) or CHP (B) for 30 min of incubation. Staphylococcal cells were collected, and total RNA was extracted. The relative transcription levels of *srrA* and *srrB* were analyzed by qRT-PCR in comparison to the expression level of *gyrB* (housekeeping gene). The experiments were performed in triplicate and repeated at least three times. Data were represented as the means ± SDs; ***, $P < 0.001$.

microaerobic conditions ($-O_2$), this time lag was reduced to 1–2 h. The growth defect of the Δ*srrAB* mutant was alleviated by the shift from oxic to microaerobic conditions, where oxygen utilization and free oxygen radical production were stringent (Fig. 2).

### Deletion of *srrAB* increased ROS accumulation during bacterial growth

ROS levels in the cultures of staphylococci were determined using a nitroblue tetrazolium (NBT) assay. The blue formazan resulting from the reduction of NBT by ROS generated from the respiratory metabolism and growth was measured spectrophotometrically, and the absorbance value ($OD_{575}$) is expected to reflect the ROS levels.

Although ROS accumulation in the bacterial cultures got lower with the initial inoculum decreased, the ROS level of the *srrAB* deletion mutant was always higher than that of SE1457 in the post-stationary phase (Fig. 3). The gap in the ROS levels between the *srrAB* deletion mutant and SE1457 was reduced by a shift from oxic condition to microaerobic condition. ROS accumulation in the cultures of the wild-type strain SE1457 remained at a relatively stable and low level (about $OD_{575}$ of 0.1) during bacterial

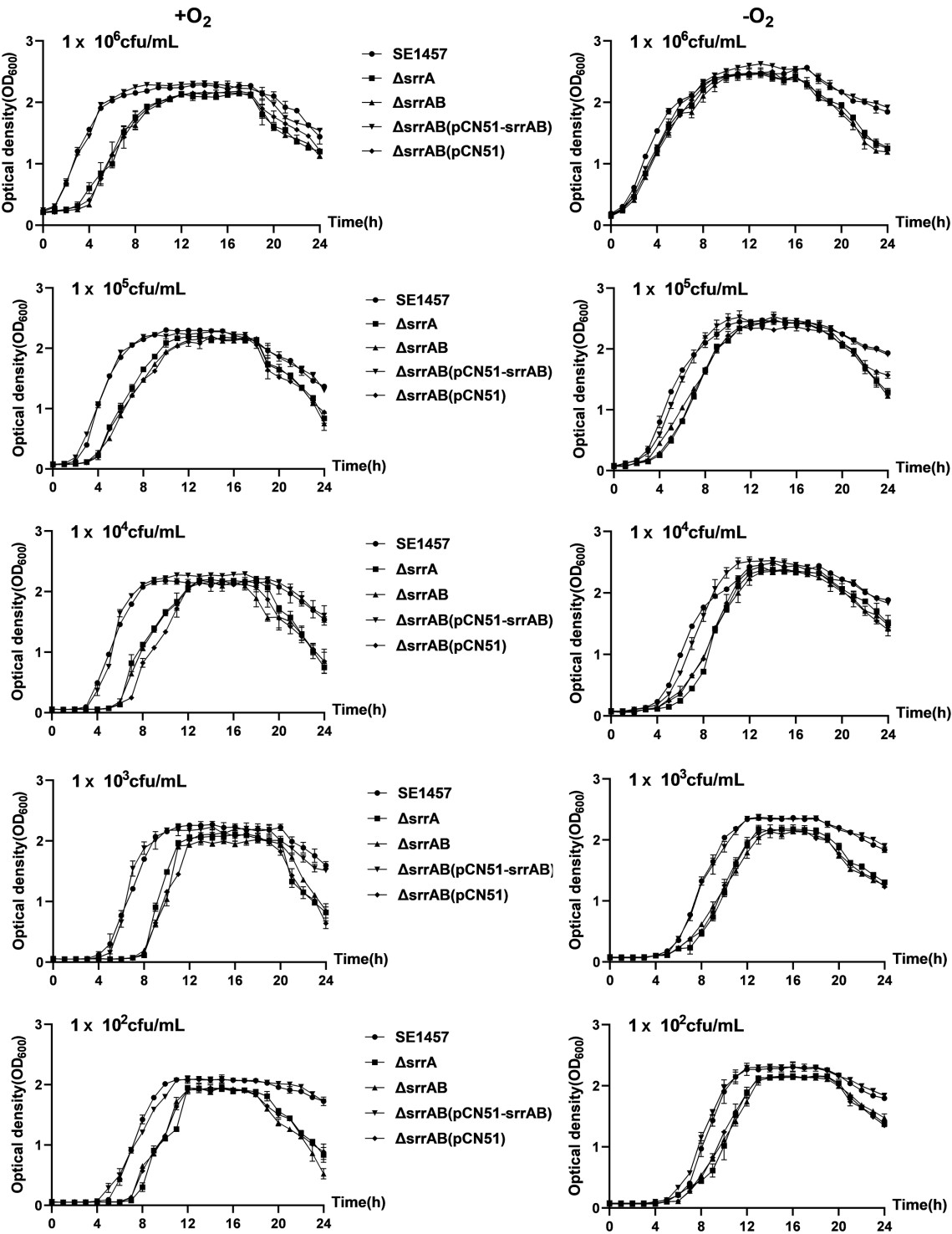

**FIG 2** Growth curves of SE1457 srrAB isogenic mutants under oxic and microaerobic conditions. Overnight cultures of SE1457, ΔsrrAB, ΔsrrAB(pCN51-srrAB), and ΔsrrAB(pCN51) strains were diluted (1:200) in fresh TSB medium and incubated at 37°C for about 4 h with the OD600 reaching 0.8. After 10-fold serial dilutions, the bacterial suspension was inoculated (1:200) into a fresh TSB medium. Under oxic conditions (+$O_2$), bacterial suspension was added to triplicate wells (200 µL/well) in a 96-well plate and placed into a heated microplate reader that allowed for free diffusion of gases. Under microaerobic conditions (−$O_2$), the cultures were inoculated into a 96-well plate completely filled with the medium, all air bubbles were removed, and the plate was sealed with sealing film. The cultures were measured hourly at an OD600 for 24 h. The curves represented the results of the three independent experiments. Data were represented as the means ± SD of triplicate wells.

growth. However, it exhibited a sharp increase (about twofold) in the ΔsrrAB and ΔsrrA mutants compared to SE1457, particularly in the decline phase of the growth curve. The complementation of srrAB into the ΔsrrAB mutant restored ROS accumulation to the wild-type level, whereas the introduction of vector control had no effect.

## Deletion of srrAB reduced resistance to oxidative stress

Tenfold serial dilutions of staphylococcus cells were spotted on tryptone soy agar (TSA) containing 0.5 mM to 1 mM of $H_2O_2$ or CHP at 37°C for 24 h of incubation. Under both oxic and microaerobic conditions, the ΔsrrAB and ΔsrrA mutants exhibited dramatically decreased resistance (100- to 1,000-fold) to $H_2O_2$ and CHP compared to SE1457. This reduced resistance was rescued by complementation with srrAB. Furthermore, the resistance of the ΔsrrAB and ΔsrrA mutants to oxidative stress was rescued by the addition of the ROS inhibitor 2,2-dipyridyl (DIP). In contrast, the colony density and size of the srrAB deletion mutant were not completely restored to the levels of the parent strain, indicating that the srrAB deletion mutant decreased the ability to detoxify ROS, and its growth rate was not entirely ruled out (Fig. 4).

Consistent with the stress plate assay results, the growth of the ΔsrrAB and ΔsrrA mutants cultured in TSB medium with 1 mM $H_2O_2$ was significantly inhibited compared to SE1457 under both oxic and microaerobic conditions (Fig. 5). When the bacterial inoculum was as low as $1 \times 10^2$ CFU/mL, the growth of the mutants was inhibited completely, while it was rescued by srrAB complementation. These results indicated that srrAB plays a vital role in cellular response to oxidative stress in S. epidermidis.

## Deletion of srrAB reduced survival in macrophages

To explore the resistance against macrophage killing, researchers conducted a bactericidal assay using mouse macrophage Ana-1 cells infected with SE1457 isogenic srrAB mutants. They evaluated the phagocytosis rate by CFU counting before lysing the Ana-1 cells. The phagocytosis rates of SE1457 and ΔsrrAB mutant were 97.79% and 98.66%, respectively. There was no significant difference in phagocytosis rate between SE1457 and ΔsrrAB ($P > 0.05$; Fig. 6A). After 6 h co-incubation, the number of the surviving bacteria recovered from ΔsrrAB-infected Ana-1 cells ($2.06 \times 10^6$ CFU/mL) were significantly lower than those from SE1457 ($5.44 \times 10^6$ CFU/mL) and ΔsrrAB(pCN51-srrAB) ($6.2 \times 10^6$ CFU/mL; $P < 0.001$). The ΔsrrA mutant and ΔsrrAB(pCN51) vector control exhibited similar levels of reduced survival (Fig. 6B).

## Deletion of srrAB increased intracellular ROS levels

To determine whether the reduced survival of the ΔsrrAB mutant in macrophages resulted from ROS accumulation, intracellular ROS production was assessed using a fluorescent dye (2',7'-dichloro-dihydro-fluorescein diacetate dye [DCFH-DA]). As expected, the numbers of the ROS-positive cells infected by ΔsrrA and ΔsrrAB were significantly increased compared to those infected by SE1457 and ΔsrrAB(pCN51-srrAB) using flow cytometry. Among 50,000 cells analyzed, both the ΔsrrA and ΔsrrAB mutants (8,988 ± 120, 19.1%; 10,708 ± 187, 22.8%, respectively) exhibited approximately a twofold increase in the number of ROS-positive cells compared to the parent strain SE1457 (4,999 ± 339, 9.9%). The complementation strain ΔsrrAB(pCN51-srrAB) (4,489 ± 123, 9.7%) restored the number of ROS-positive cells to the wild-type level, while the vector control ΔsrrAB(pCN51) (8,937 ± 334, 18.9%) mirrored the ΔsrrAB mutant ($P < 0.01$; Fig. 7A; Table S1). Consistently, fluorescence intensity assessed by fluorescence spectrophotometry indicated higher fluorescence intensity in Ana-1 cells infected with the ΔsrrA and ΔsrrAB mutants, as well as the vector control ΔsrrAB(pCN51), compared to those infected with the parent strain SE1457 and complementation strain ΔsrrAB(pCN51-srrAB) ($P < 0.01$; Fig. 7B). These results suggest that SrrAB mitigates ROS to protect S. epidermidis from phagocytic clearance.

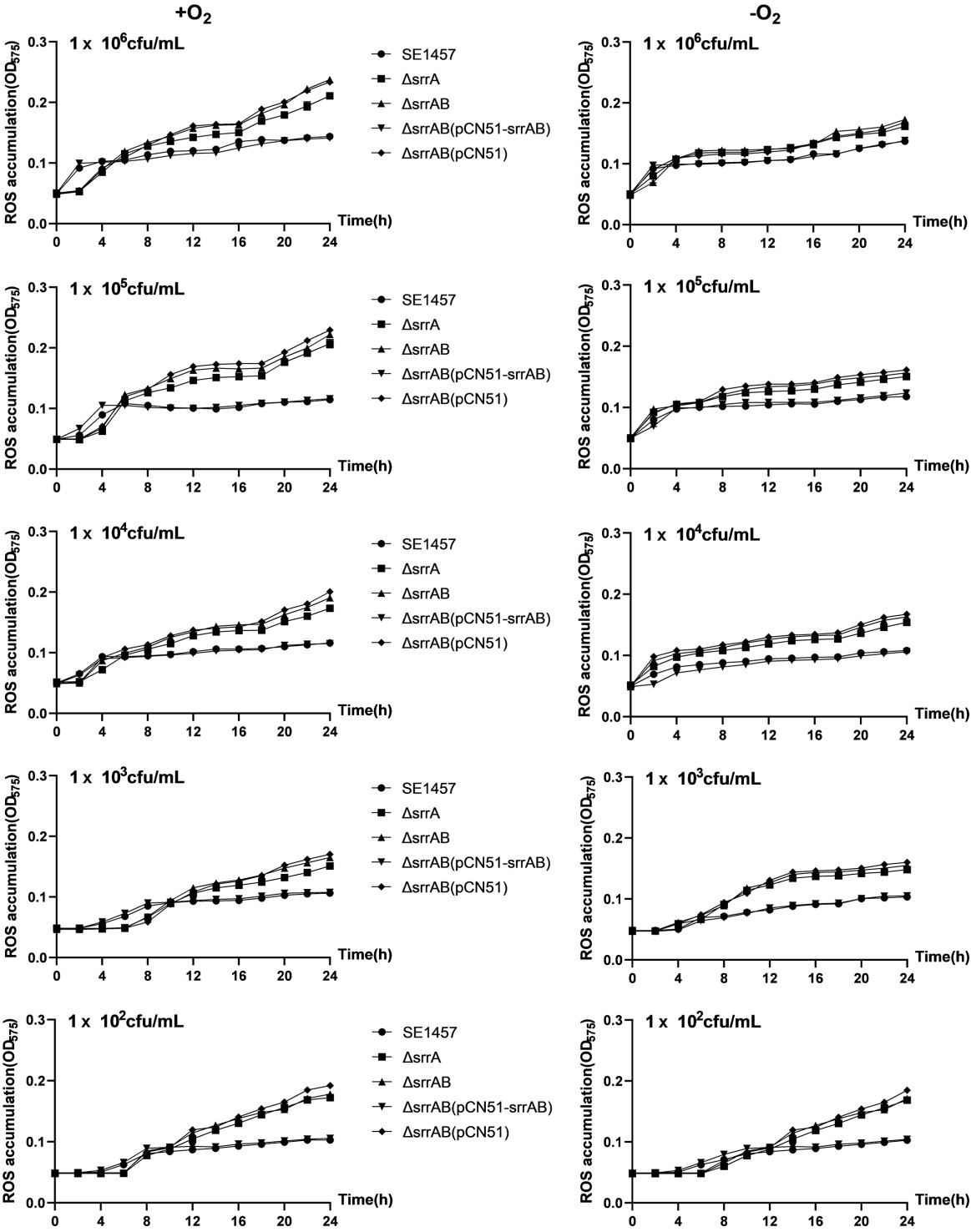

**FIG 3** ROS accumulation of SE1457 *srrAB* isogenic mutants under oxic and microaerobic conditions. Overnight cultures were diluted (1:200) into fresh TSB medium and incubated at 37°C with aeration for 4 h ($OD_{600}$ of 0.8). The bacterial suspension was adjusted to $1.0 \times 10^6$ CFU/mL and serially diluted (10-fold), then pipetted into the microplate. At each time point, 100 µL of bacterial suspension was withdrawn, and 0.5 mL NBT (1 mg/mL) was added. The cultures were measured every 2 h at an $OD_{575}$ for 24 h. The experiments were repeated at least three times, and one representative result was shown. Data were represented as means ± SD of triplicate wells.

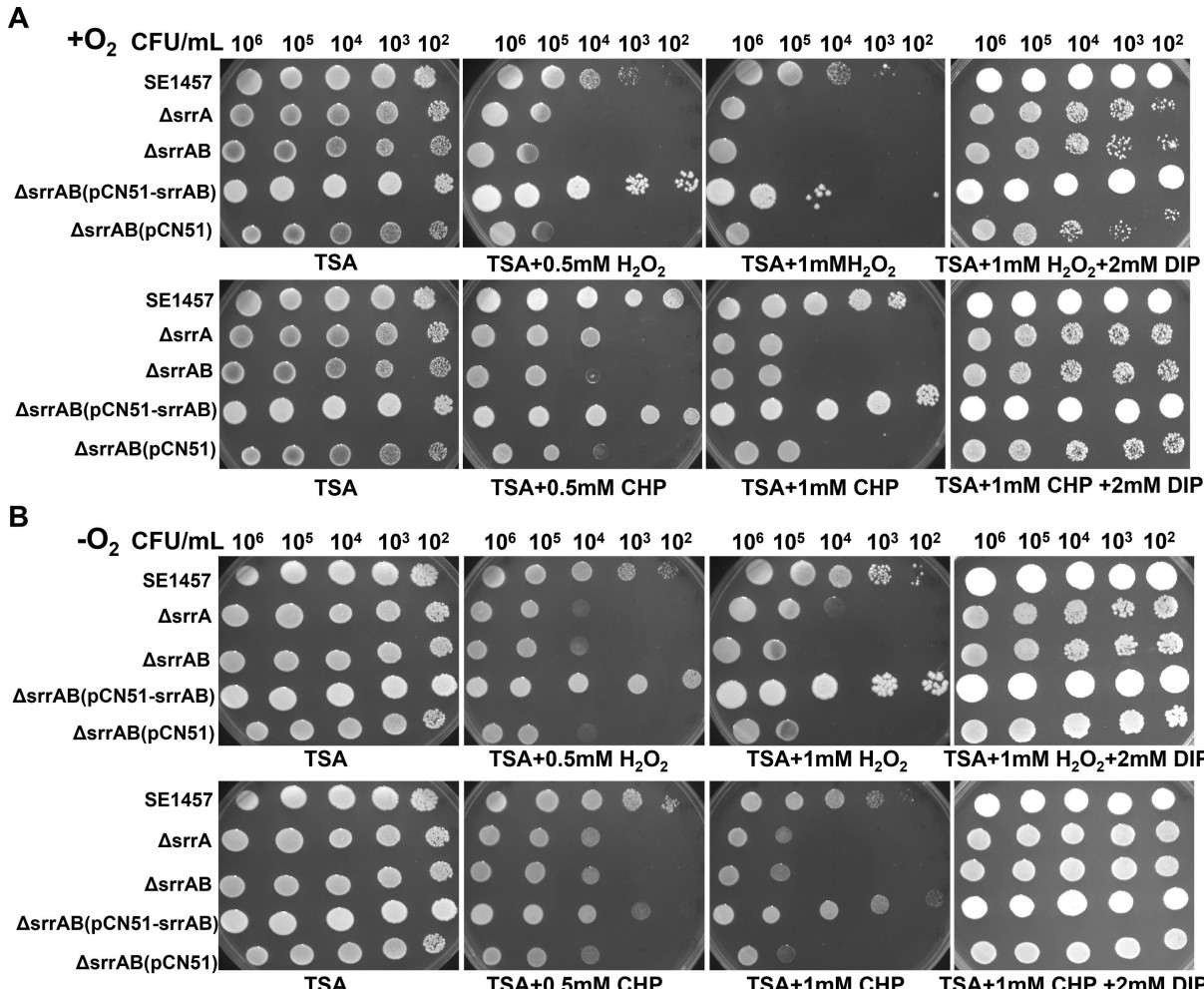

**FIG 4** Tolerance of the Δ*srrAB* mutant to oxidative stress under oxic and microaerobic conditions. Staphylococci grown to OD$_{600}$ value of 0.8 at 37°C with aeration for about 4 h were serially diluted (1:10), and an aliquot (5 µL) was spotted onto the TSA plate containing different concentrations of H$_2$O$_2$ or CHP. For the detoxification of ROS, DIP was added. The plates were incubated at 37°C for 24 h under oxic conditions (+O$_2$) or under microaerobic conditions (−O$_2$). The image labeled "TSA" on the lower left in panel A or B was the duplicate serving as negative control from a set of assays. The results represented one of three independent experiments.

## Transcriptional profile comparison of Δ*srrAB* and SE1457

RNA-seq was performed to compare the transcriptional profile of Δ*srrAB* with that of SE1457. More than 90% of the reads mapped to *S. epidermidis* RP62A after removal of ambiguous nucleotides. It revealed 610 differentially expressed genes, encompassing pathways related to oxidative stress, respiratory and energy metabolism, transition ion homeostasis, biofilm formation, and DNA replication, transcription, and translation. Among these, 490 genes were upregulated, and 120 were downregulated in the Δ*srrAB* mutant. Downregulated genes included those involved in ROS-scavenging and environmental adaptation (*scdA* [iron-sulfur cluster repair protein], *serp1797* [NAD-dependent protein deacetylase], and *serp0483* [thioredoxin]), respiratory and energy metabolism (*nrdDG* [anaerobic ribonucleoside-triphosphate reductase] and *serp2330* [thiamine VitB1 biosynthesis protein]), tricarboxylic acid cycle (TCA) cycle (*pflA* [pyruvate formate-lyase-activating enzyme] and *pflB* [formate acetyltransferase]), and biofilm and persister formation (*icaA* [poly-beta-1,6-N-acetyl-D-glucosamine synthase] and *serp1681* [endoribonuclease MazF]), among others. However, the transcriptional levels of other antioxidant genes such as *sodA* (superoxide dismutase), *serp1478* (Dps analog), *serp1398* (PerR

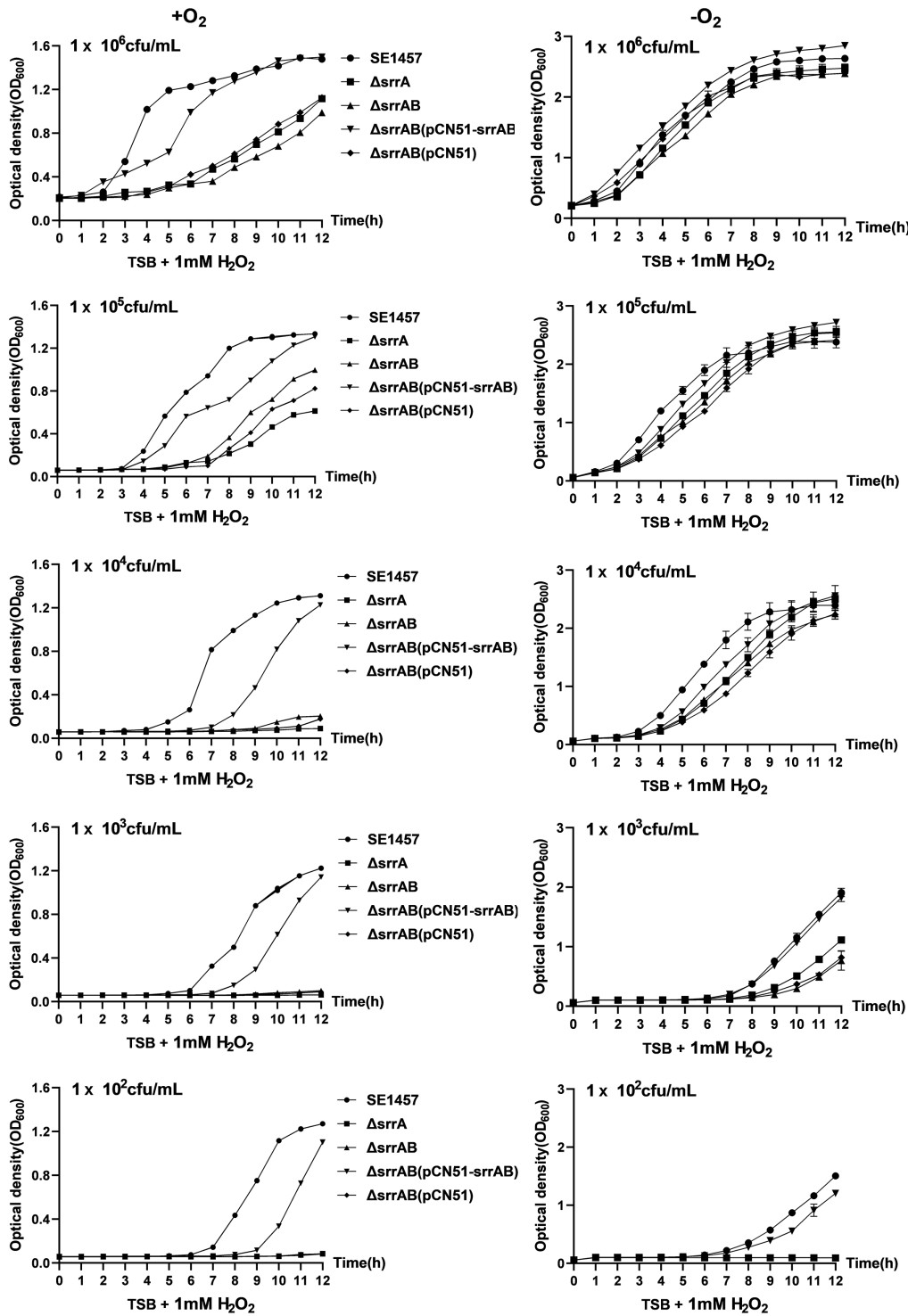

**FIG 5** Growth curves of SE1457 *srrAB* isogenic mutants under oxidative stress. Staphylococci grown to $OD_{600}$ value of 0.8 at 37°C with shaking were 10-fold serially diluted into TSB medium containing 1 mM $H_2O_2$. The incubation under oxic ($+O_2$) and microaerobic ($-O_2$) conditions were the same as described in the previous experiments. The cultures were measured hourly at an $OD_{600}$ for 12 h. The curves represented the results of one of the three independent experiments. Data were represented as the means ± SD of triplicate wells.

analog), *abfR* (MgrA analog), and *SarA* (transcriptional regulator) remained unchanged (Table 1; Table S2).

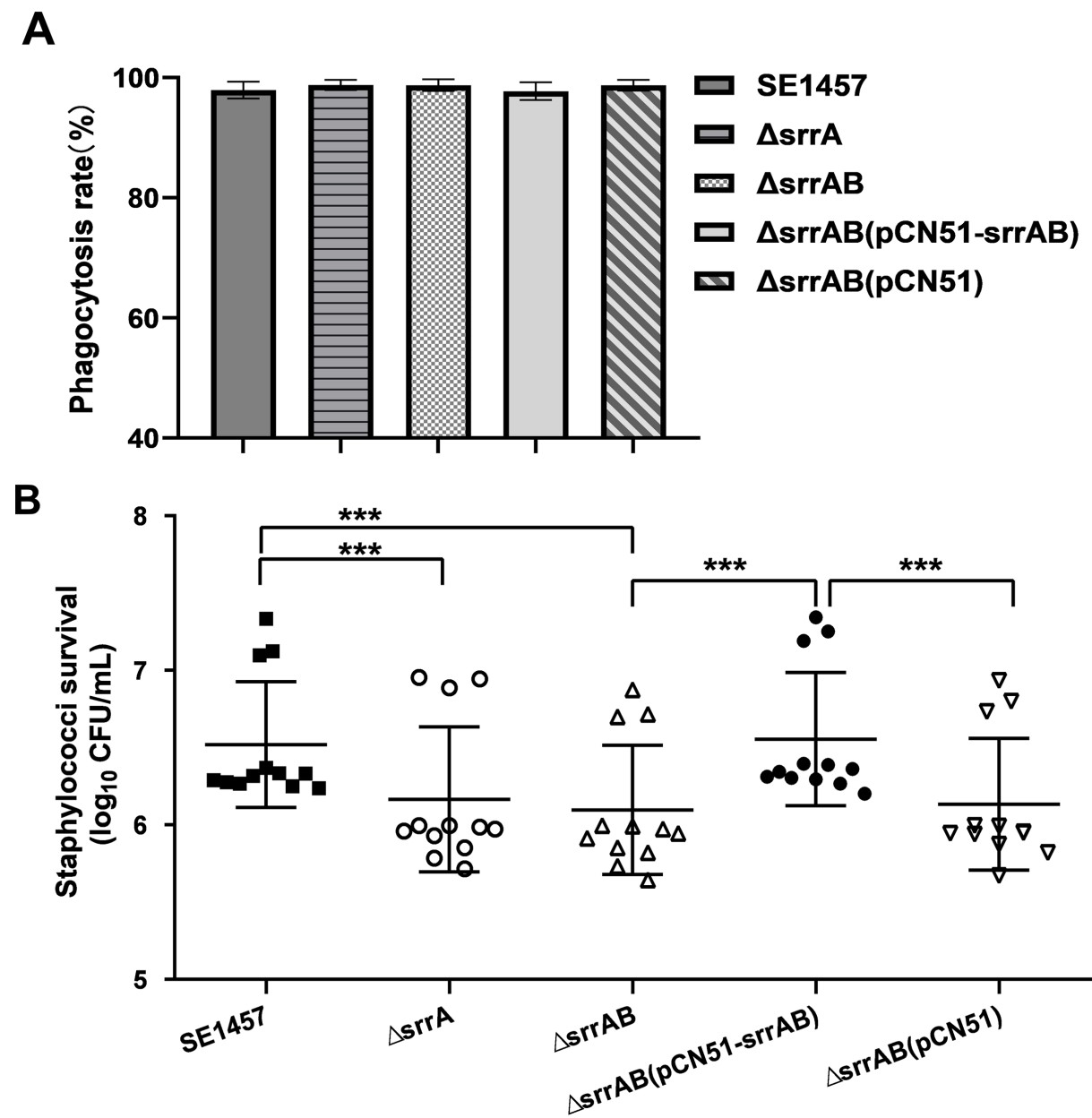

**FIG 6** Effect of *srrAB* deletion on the phagocytosis of *S. epidermidis* by macrophages. Ana-1 cells ($1.5 \times 10^7$ cells/mL, 1 mL) were co-incubated with *S. epidermidis* strains ($1.5 \times 10^8$ CFU/mL, 1 mL) at a multiplicity of infection of 10 in a six-well plate at 37°C with 5% $CO_2$. At the time point of 6 h incubation, the extracellular bacterial cells unengulfed by macrophages were harvested, and the phagocytosis rate of *S. epidermidis* by macrophage was evaluated by CFU counting prior to Ana-1 cell lysis. Statistical significance was determined by one-way analysis of variance (ANOVA) analysis ($P > 0.05$) (A). The Ana-1 cells were treated with 100 µg/mL gentamycin and 20 µg/mL lysostaphin for 30 min to kill the extracellular and adherent bacterial cells and lysed with radioimmunoprecipitation assay buffer (RIPA) lysis buffer. The survival of *S. epidermidis* strains was assessed by CFU counting on TSA plate (B). The experiments were repeated at least three times, and the data were represented as the means ± SD. ***, $P < 0.001$. The black square and circle represent the parent strain SE1457 and complementation strain Δ*srrAB*(pCN51-*srrAB*), respectively. The white circle represents the Δ*srrA* mutant. The white triangle and inverted triangle represent the Δ*srrAB* mutant and Δ*srrAB*(pCN51) vector control, respectively.

## Validation of the sequencing data by qRT-PCR

To validate the sequencing data, researchers further verified the transcriptional levels of genes related to phenotypic changes, particularly those involved in the oxidative response, by qRT-PCR without $H_2O_2$ treatment. Of the 19 selected genes, 17 expressions were well consistent with the RNA-seq data, as shown in Table 1. In addition, two genes

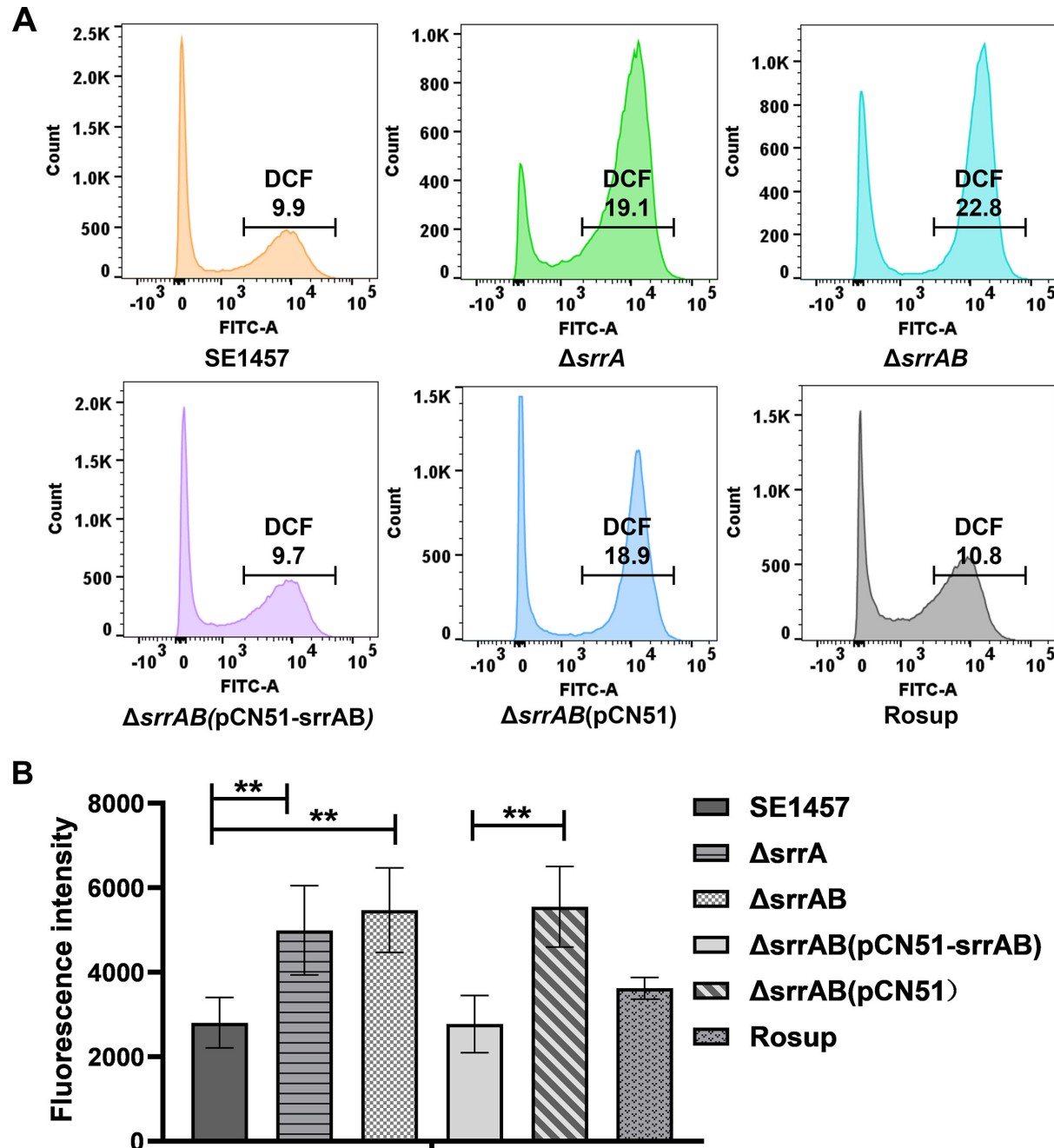

**FIG 7** Effect of *srrAB* deletion on the ROS accumulation of macrophages infected by *S. epidermidis*. Mouse macrophage Ana-1 cells were co-incubated with *S. epidermidis* strains at a multiplicity of infection (MOI) of 10 in a six-well plate at 37°C for 6 h with 5% $CO_2$. Samples were stained with DCFH-DA (10 μmol/L) at 37°C for 20 min in the dark, and the infected cells were washed three times with serum-free RPMI-1640 medium. The ROS-positive cells and fluorescence intensity were measured by flow cytometry (A) and a Microplate Reader (B), respectively. Rosup was designated as a positive control for the induction of ROS. The figures represent one of three independent experiments, and the data were represented as the means ± SD. **, $P < 0.01$.

(*katA* and *ahpC*) non-differentially expressed in RNA-seq analysis showed differential expression, which was inconsistent with the RNA-seq. Collectively, the sequencing results were reliable.

qRT-PCR further determined transcriptional levels of ROS-scavenging genes under oxidative stress, which showed downregulation of genes *katA*, *ahpC*, *scdA*, *serp1797*, and *serp0483* (about 25-, 33-, 100-, 50-, and 100-fold) in the Δ*srrAB* mutant challenged with $H_2O_2$, respectively. Conversely, *srrAB* complementation led to upregulation of these

**TABLE 1** Genes differentially expressed between the Δ*srrAB* mutant and the parent strain SE1457 under aerobic conditions

| Gene and locus | GenBank accession no. (location) | Description or predicted function | Expression ratio (mutant/WT) | |
|---|---|---|---|---|
| | | | RNA-seq | qRT-PCR[a] |
| Oxidative stress and environmental stress | | | | |
| katA | CP000029.1:914553–916067 | Catalase | 0.75 | 0.26 ± 0.05 |
| ahpC | CP000029.1:47868–48437 | Alkyl hydroperoxide reductase, subunit C | 0.81 | 0.19 ± 0.07 |
| scdA | CP000029.1:328812–329486 | Iron-sulfur cluster repair protein ScdA | 0.20 | 0.2 ± 0.03 |
| SERP0480 | CP000029.1:476297–476719 | Organic hydroperoxide resistance protein | 0.45 | ND |
| recF | CP000029.1:2611754–2612869 | DNA replication and repair protein RecF | 0.47 | 0.2 ± 0.16 |
| SERP0815 | CP000029.1:818735–819607 | DNA processing protein DprA, putative | 0.40 | 0.15 ± 0.12 |
| SERP1797 | CP000029.1:1842789–1843529 | NAD-dependent protein deacetylase | 0.44 | 0.19 ± 0.05 |
| SERP0483 | CP000029.1:478402–478725 | Thioredoxin, putative | 0.44 | 0.2 ± 0.09 |
| SERP0257 | CP000029.1:265167–266189 | Alcohol dehydrogenase, zinc-containing | 0.21 | ND |
| SERP0760 | CP000029.1:759097–759894 | Glyoxalase family protein | 0.14 | 0.1 ± 0.09 |
| SERP2080 | CP000029.1:2106578–2108017 | Aldehyde dehydrogenase family protein | 0.38 | ND |
| SERP2326 | CP000029.1:2362671–2363624 | TPP[b]-dependent acetoin dehydrogenase E1 alpha-subunit | 0.38 | ND |
| SERP1286 | CP000029.1:1330053–1330493 | Organic hydroperoxide resistance protein | 0.43 | ND |
| SERP1273 | CP000029.1:1318052–1318552 | Universal stress protein family | 0.20 | 0.17 ± 0.12 |
| SERP0847 | CP000029.1:858313–859017 | Oxidoreductase, short-chain dehydrogenase/reductase family | 4.21 | ND |
| fhs | CP000029.1:1341346–1343013 | Formate—tetrahydrofolate ligase | 11.93 | 3.07 ± 0.9 |
| SERP1765 | CP000029.1:1808640–1809707 | Iron-sulfur cluster carrier protein | 2.67 | ND |
| SERP0972 | CP000029.1:990579–990812 | Cold shock protein CspC | 3.31 | ND |
| lrgB | CP000029.1:2047486–2048187 | Antiholin-like protein LrgB | 2.28 | ND |
| sodA | CP000029.1:1158272–1158871 | Superoxide dismutase | 0.90 | 0.59 ± 0.14 |
| Respiratory chain and energy metabolism | | | | |
| nrdG | CP000029.1:2215128–2215664 | Anaerobic ribonucleoside-triphosphate reductase-activating protein | 0.16 | 0.02 ± 0.01 |
| nrdD | CP000029.1:2215661–2217511 | Anaerobic ribonucleoside-triphosphate reductase | 0.18 | 0.06 ± 0.07 |
| qoxA | CP000029.1:638799–640787 | Quinol oxidase subunit 1 | 0.16 | ND |
| qoxB | CP000029.1:640787–641911 | Quinol oxidase subunit 2 | 0.14 | ND |
| qoxC | CP000029.1:638204–638809 | Quinol oxidase subunit 3 | 0.16 | ND |
| qoxD | CP000029.1:637917–638207 | Quinol oxidase subunit 4 | 0.15 | ND |
| SERP2330 | CP000029.1:2367078–2367989 | Thiamine (VitB1） biosynthesis protein | 0.44 | ND |
| SERP1773 | CP000029.1:1816844–1817158 | Heme-degrading monooxygenase | 0.25 | ND |
| mqo-3 | CP000029.1:2350001–2351497 | Malate:quinone-oxidoreductase，MQO | 50.76 | ND |
| mqo-1 | CP000029.1:1970556–1972034 | Malate:quinone-oxidoreductase | 2.40 | ND |
| hemC | CP000029.1:1269281–1270207 | Porphobilinogen deaminase | 2.74 | ND |
| TCS | | | | |
| srrA | CP000029.1:1101714–1102439 | DNA-binding response regulator ResD | 0.00 | 0.00001 |
| srrB | CP000029.1:1099964–1101733 | Sensor histidine kinase ResE | 0.00 | 0.00001 |
| SERP1895 | CP000029.1:1917426–1918115 | Transcriptional regulator, DeoR family | 0.46 | ND |
| SERP0635 | CP000029.1:626997–627416 | Transcriptional regulator, MarR family | 0.30 | ND |
| saeS | CP000029.1:364454–365419 | Sensor histidine kinase | 26.88 | ND |
| saeR | CP000029.1:365509–366198 | Response regulator SaeR | 28.53 | ND |
| vraS | CP000029.1:1485358–1486404 | Sensor histidine kinase | 2.60 | ND |
| vraR | CP000029.1:1484739–1485368 | DNA-binding response regulator | 3.03 | ND |
| cadC | CP000029.1:2258429–2258776 | Transcriptional regulator, ArsR family | 6.04 | ND |
| codY | CP000029.1:826348–827118 | Transcriptional regulator CodY | 2.42 | 2.03 ± 0.29 |
| TCA | | | | |
| pflA | CP000029.1:2411378–2412133 | Pyruvate formate-lyase-activating enzyme | 0.09 | 0.1 ± 0.07 |
| pflB | CP000029.1:2412155–2414401 | Formate acetyltransferase | 0.11 | 0.1 ± 0.05 |
| SERP2324 | CP000029.1:2360269–2361546 | Dihydrolipoamide acetyltransferase | 0.25 | ND |

*(Continued on next page)*

**TABLE 1** Genes differentially expressed between the Δ*srrAB* mutant and the parent strain SE1457 under aerobic conditions (*Continued*)

| Gene and locus | GenBank accession no. (location) | Description or predicted function | Expression ratio (mutant/WT) | |
| --- | --- | --- | --- | --- |
| | | | RNA-seq | qRT-PCR[a] |
| SERP2169 | CP000029.1:2198607–2199425 | Putative pyruvate, phosphate dikinase regulatory protein 2 | 0.02 | ND |
| ppdK | CP000029.1:2199427–2202054 | Pyruvate phosphate dikinase | 0.04 | ND |
| SERP2147 | CP000029.1:2173685–2174401 | Hypothetical cytosolic protein | 9.62 | ND |
| ptsI | CP000029.1:665150–666868 | Phosphoenolpyruvate-protein phosphotransferase | 3.02 | ND |
| pyc | CP000029.1:699164–702607 | Pyruvate carboxylase | 2.36 | ND |
| Biofilm and persistent bacteria | | | | |
| icaA | CP000029.1:2334220–2335458 | Poly-beta-1,6-N-acetyl-D-glucosamine synthase | 0.30 | ND |
| icaB | CP000029.1:2335724–2336593 | Polysaccharide deacetylase family protein | 0.40 | ND |
| SERP1681 | CP000029.1:1725803–1726165 | Endoribonuclease MazF | 13.44 | ND |
| SarA | CP000029.1:279424–279798 | Transcriptional regulator SarA | 1.90 | 2.32 ± 1.16 |
| Maintaining metal ion homeostasis | | | | |
| SERP1120 | CP000029.1:1159276–1159695 | Transcriptional regulator, Fur family | 1.51 | ND |
| cysI | CP000029.1:2222684–2224402 | Sulfite reductase [NADPH] hemoprotein beta-component | 1.87 | ND |
| SERP1978 | CP000029.1:1995611–1996240 | Nitroreductase family protein | 2.35 | ND |
| SERP1039 | CP000029.1:1084763–1085488 | Menaquinone biosynthesis methyltransferase, putative | 1.32 | ND |
| SERP1777 | CP000029.1:1821012–1822007 | Ferric citrate ABC transporter periplasmic binding protein | 1.63 | ND |
| mnhB | CP000029.1:520098–520526 | Monovalent cation/proton antiporter | 1.34 | ND |
| sitB | CP000029.1:293956–294792 | Iron-chelated ABC transporter | 1.25 | ND |
| isdG | CP000029.1:1816844–1817158 | Heme-degrading monooxygenase | 1.25 | ND |
| Oxidoreductase | | | | |
| gpxA-1 | CP000029.1:886540–887016 | Glutathione peroxidase | 0.71 | ND |
| gpxA-2 | CP000029.1:2227609–2228085 | Glutathione peroxidase | 0.17 | ND |
| SERP2245 | CP000029.1:2272878–2273636 | S-formylglutathione hydrolase FrmB | 0.15 | ND |
| dep | CP000029.1:2127505–2129109 | Gamma glutamyl transpeptidase | 2.92 | ND |
| SERP2195 | CP000029.1:2228098–2229441 | Dihydrolipoamide dehydrogenase | 0.48 | ND |
| gnd | CP000029.1:1116266–1117672 | 6-Phosphogluconate dehydrogenase, decarboxylating | 1.64 | ND |
| Phosphorylation | | | | |
| atpC | CP000029.1:1748909–1749313 | ATP synthase epsilon chain | 0.008 | ND |
| purQ | CP000029.1:646768–647439 | Phosphoribosylformylglycinamidine synthase subunit PurQ | 4.72 | ND |
| purL | CP000029.1:647432–649621 | Phosphoribosylformylglycinamidine synthase subunit PurL | 4.28 | ND |
| purC | CP000029.1:651077–652108 | Phosphoribosylformylglycinamidine cyclo-ligase | 3.99 | ND |
| purK | CP000029.1:644674–645801 | N5-carboxyaminoimidazole ribonucleotide synthase | 3.85 | ND |
| pyrG | CP000029.1:1772905–1774512 | CTP synthase | 3.44 | ND |

[a]qRT-PCR data are given as the means ± SDs of the results from three independent experiment. ND, not done.
[b]TPP, thiamine pyrophosphate.

genes (3.9-, 3-, 3.5-, 3.9-, and 2.7-fold, respectively), while the Δ*srrA* mutant and vector control mirrored the Δ*srrAB* mutant (Fig. 8).

## Binding of recombinant SrrA protein to putative promoter regions

To further elucidate the regulatory role of SrrAB in oxidative stress, an electrophoretic mobility shift assay (EMSA) was performed to detect the binding of recombinant SrrA (His-tagged SrrA) to putative promoter regions labeled with biotin. The 269 bp DNA fragment upstream of *srrAB* (p-*srrAB*) formed a shifted complex with phosphorylated SrrA (SrrA-P) in a dose-dependent manner (Fig. 9A, lanes 2–6). Specific competition with

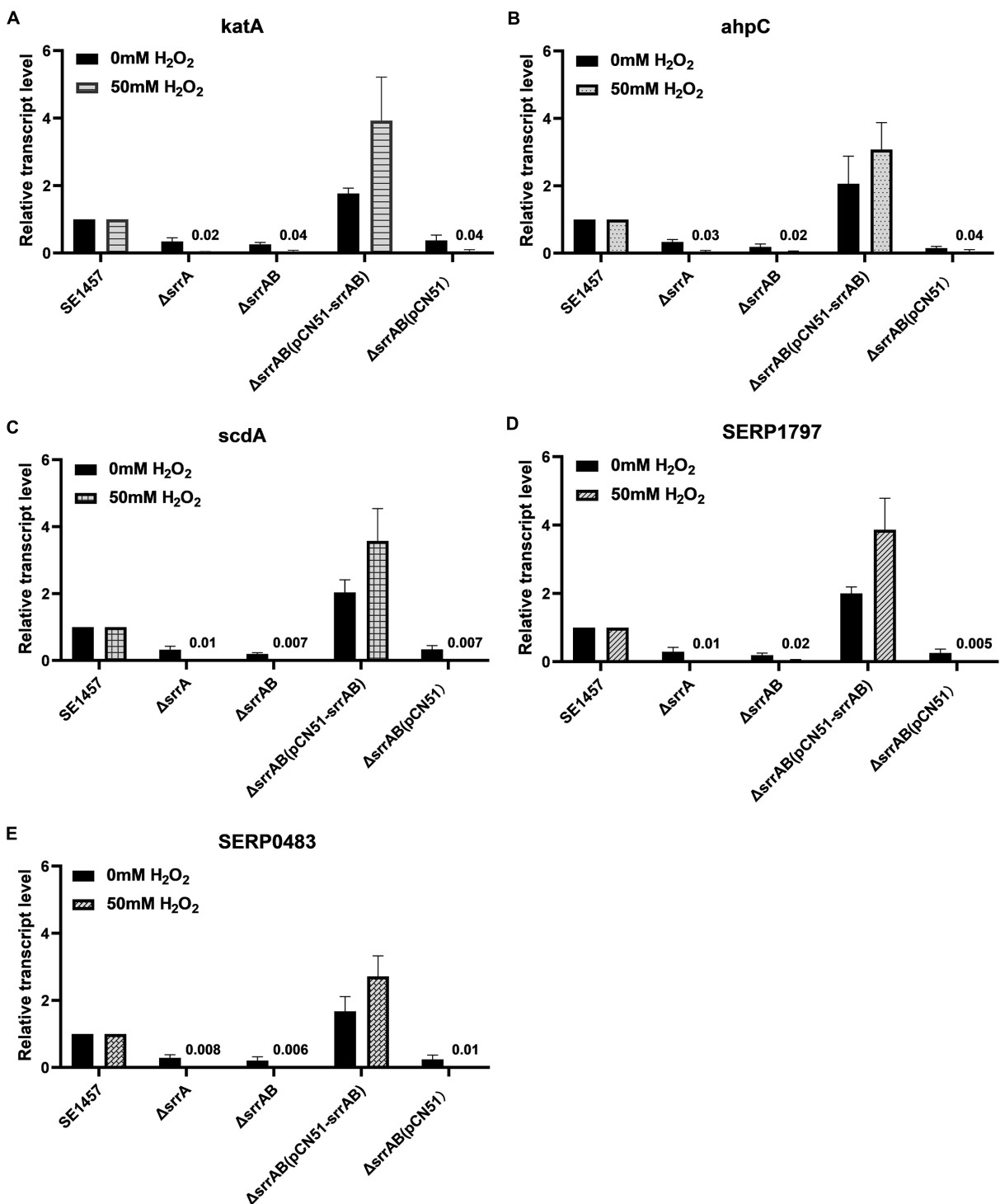

**FIG 8** Transcriptional levels of ROS-scavenging genes in SE1457 *srrAB* isogenic mutants. *S. epidermidis* strains were inoculated into TSA medium challenged with or without 50 mM $H_2O_2$ and incubated at 37°C for 6 h. Bacterial cells were harvested, and total RNA was extracted. The relative expression levels of *katA* (A), *ahpC* (B), *scdA* (C), *serp1797* (D), and *serp0483* (E) were analyzed by qRT-PCR in comparison to the transcription level of *gyrB* (housekeeping gene). The measured values located on the bar were represented as the means of the relative transcript level. The experiments were performed in triplicate and repeated at least three times. Data were represented as the means ± SD.

125-fold excess unlabeled p-*srrAB* inhibited SrrA-DNA-probe complex formation (Fig. 9A, lane 7). In contrast, the same amount of unlabeled nonspecific DNA (258 bp fragment of *gyrB* coding region) had no effect (Fig. 9A, lane 8). SrrA-P also caused a mobility shift of

the 248 bp, 303 bp, 291 bp, 337 bp, or 303 bp fragments upstream of *katA*, *ahpC*, *scdA*, *serp1797*, or *serp0483*, respectively (Fig. 9B through F), but not a 283 bp DNA fragment of the *rpsJ* gene that served as a negative control (Fig. 9G). The results showed that SrrA-P could bind specifically to the promoter regions of certain oxidative stress genes.

Furthermore, our previous work found that SrrA-P could bind to the putative promoter regions of *icaA*, *icaR*, *qoxB*, *pflAB*, *ctaA*, *altE*, and *serp1281* (27). A motif-based sequence analysis revealed a conserved SrrA binding box C(T)T(A)CCTCCT or AGGAGGA(T)G(A) as the reverse complement pattern, which was present in the promoter regions of all these target genes (Fig. 10).

## DISCUSSION

Oxidation-sensing regulators have been extensively investigated in pathogenic bacteria, including *S. aureus* (29–32), *Escherichia coli* (33–35), *Pseudomonas aeruginosa* (36, 37), and *Mycobacterium tuberculosis* (38, 39). The redox-sensitive regulatory proteins act as major regulators of bacteria adaptability and resistance to oxidative stress. As a staphylococcal respiratory response system, much attention has been focused on the relevance of SrrAB with virulence factors, respiratory metabolism, growth, biofilm formation, and programmed cell death. In response to oxygen tension in *Staphylococcus* species, the roles of SrrAB in resistance to oxidative stress in response to ROS are mainly unknown, particularly between under oxic and anaerobic conditions in *S. aureus* and *S. epidermidis*. This study has found that *S. epidermidis* utilizes SrrAB to sense and respond to oxidative stress to increase bacterial resistance and intracellular survival upon ROS.

A previous study (27) found that *S. epidermidis* SrrAB responds to microaerobic stress selectively. In this study, the *srrA* and *srrB* genes of SE1457 were induced by the $H_2O_2$ and CHP with an exposure time of 30 min (Fig. 1). This indicated that *S. epidermidis* SrrAB responds not only to low oxygen pressure but also to oxidants. To study the role of SrrAB in regulating resistance to oxidants, researchers constructed an *srrAB* deletion mutant (Δ*srrAB*) from SE1457 (Fig. S1). Several studies (26, 28) have found that the Δ*srrA* mutant of *S. aureus* (*S. aureus* strain MW2) increased the resistance to the $H_2O_2$, and the expressions of *katA* and *dps* in the Δ*srrA* mutant were significantly higher than those in the WT. Whereas the Δ*srrAB* mutant of *S. aureus* (USA300-LAC strain) decreased the resistance to the $H_2O_2$, and the transcriptions of genes *kat*, *ahpC*, and *dps* in the Δ*srrAB* mutant were decreased compared to the parent strain. Meanwhile, SrrAB is not required to induce the transcription of these genes in cells challenged with $H_2O_2$. Partly consistent with our results, the transcription of these $H_2O_2$ scavenging genes (*katA*, *ahpC*, and *scdA*) in the Δ*srrAB* mutant decreased significantly compared to that in the parent strain SE1457 both in the absence and presence of $H_2O_2$ stress. Similar results occurred in the Δ*srrAB* mutant compared to that in the complementation strain Δ*srrAB*(pCN51-*srrAB*). In addition, the fold changes in transcription levels of those genes (Δ*srrAB*/WT or Δ*srrAB*/Δ*srrAB*[pCN51-*srrAB*]) were augmented (6.5-, 6.2-, and 20-fold, respectively) when challenged with $H_2O_2$. These results indicated that the expression of the $H_2O_2$ scavenging genes is in a *srrAB*-dependent manner in *S. epidermidis* (Fig. 8).

Although the amino acid sequence of SrrAB in *S. epidermidis* shares a high identity with that in *S. aureus*, it cannot produce carotenoid pigment and possesses only SodA superoxidase dismutase (13, 40). This indicates that the SrrAB in *S. epidermidis* modulating ROS resistance is different from that in *S. aureus*, as well as the difference in biofilm formation and growth metabolism between both species.

Like other human pathogens, *S. epidermidis* must deal with ROS derived from the normal course of aerobic metabolism or the respiratory burst of phagocytes. The growth of the Δ*srrAB* mutant was impaired compared to the SE1457 under the oxic and microaerobic conditions, as reported in a previous study (27). Moreover, the impaired growth of the Δ*srrAB* mutant was alleviated under microaerobic conditions, demonstrating that the retarded growth of the *srrAB* deletion mutant is not only due to the downregulation of growth factors (*qoxBACD*, *ctaA*, *nrdDG*, and *pflBA*) but also to inevitable byproducts from the aerobic respiratory flux, such as oxidants $H_2O_2$, $O_2^-$,

and OH⁻, which inhibit bacterial growth (Fig. 2). As expected, the stress plate assay showed that the deletion of *srrAB* led to decreased resistance against $H_2O_2$ and CHP both under oxic and microaerobic conditions, and the resistance of the Δ*srrAB* mutant challenged with oxidative stress was partly relieved when incubated under microaerobic conditions. Furthermore, the resistance of the *srrAB* deletion mutant upon oxidative stress was rescued, while the ROS inhibitor was used in the TSA plate (Fig. 4). The growth curves under oxidative stress supported these findings. The Δ*srrAB* mutant under oxic conditions hardly grew in a TSB medium containing 1 mM $H_2O_2$ when the inoculum amount reached $10^4$ CFU/mL, whereas, under microaerobic conditions, they can grow sufficiently with the same inoculum and the stress. However, they grew more slowly than that of the SE1457 parent strain (Fig. 5). We speculated that the deletion of

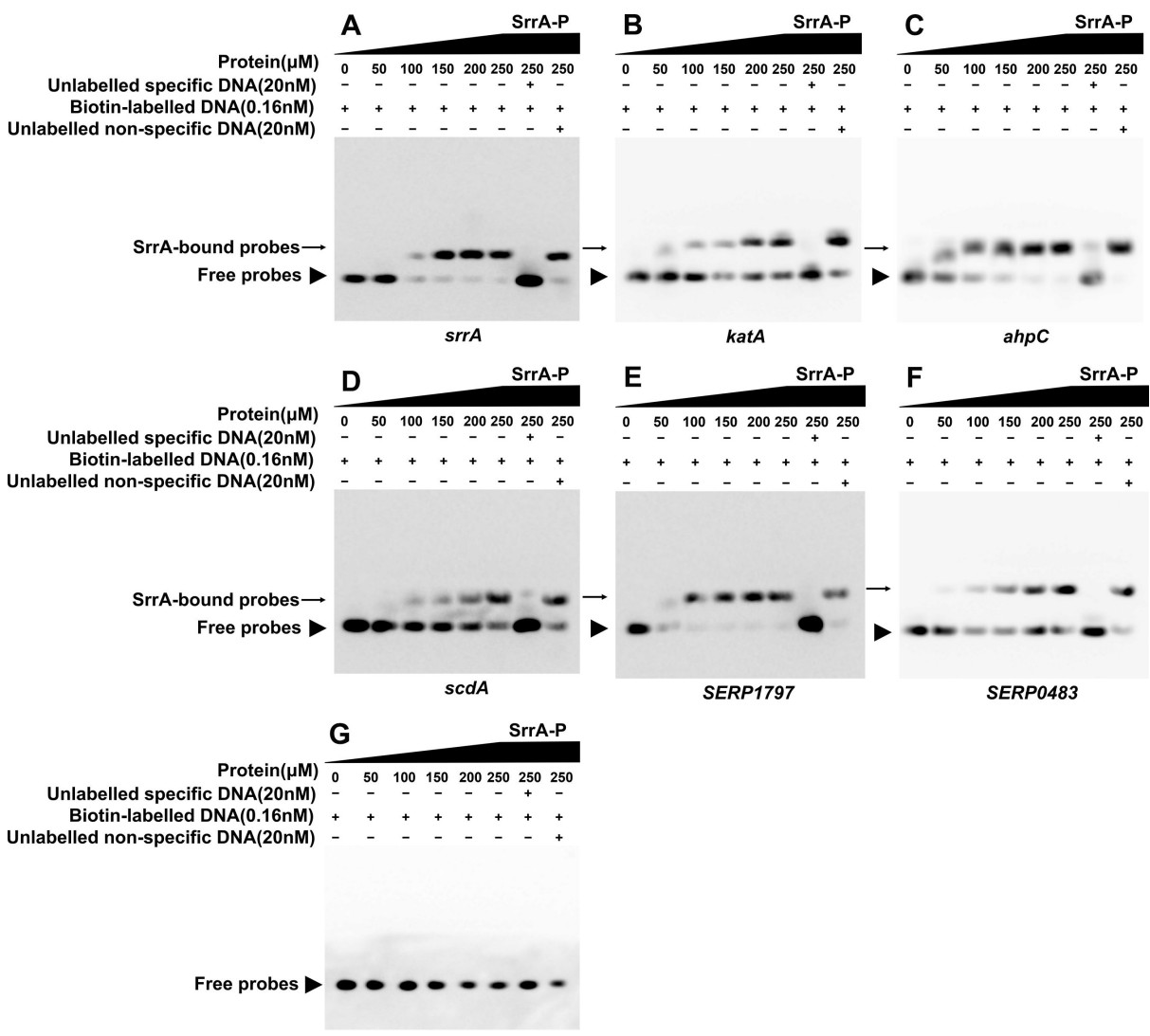

**FIG 9** EMSA analysis of *S. epidermidis* SrrA with the putative promoter regions. His-tagged SrrA was purified and phosphorylated (SrrA-P) by incubation with 50 mM acetyl phosphate. The putative promoter regions of *srrA* (A), *katA* (B), *ahpC* (C), *scdA* (D), *SERP1797* (E), and *SERP0483* (F) genes and negative control rpsJ gene (G) were PCR amplified, and the biotin-labeled DNA probes were purified. EMSAs were performed by incubating labeled probes with increasing amounts of SrrA-P. For each blot, lane 1 contained a no-protein control, and lanes 7 and 8 contained a 125-fold excess of the unlabeled specific probe (competitor control) and unlabeled nonspecific probe (DNA fragment within the *gyrB* coding region), respectively. Protein-DNA reactions were incubated at 25°C for 30 min, separated in a 6% nondenaturing polyacrylamide gel, and then blotted onto the nylon membrane. The biotin end-labeled DNA probe was detected using streptavidin conjugated to horseradish peroxidase (HRP) and a chemiluminescent substrate. Arrows indicate the position of the SrrA-DNA complex; triangles indicate the positions of free probes.

## A

| Gene/Locus | Strand | p-value | SrrA box (location) | Coding region |
|---|---|---|---|---|
| *katA* | - | 2.30e-6 | CTCCCCCT (-14 ~ -7 nt)… | ATGTCAAAAC… |
| *serp1281* | - | 2.30e-6 | CTCCTCCT (-14 ~ -7 nt)… | ATGGTGTTAT… |
| *icaR* | + | 2.30e-6 | CTCCCCCT (-17 ~ -10 nt)… | TTGAAAGATA… |
| *pflBA* | - | 6.75e-6 | CTCCGCCT (-11 ~ -4 nt)… | ATGTTAGAAA… |
| *scdA* | - | 1.43e-5 | TTCCTCCT (-13 ~ -6 nt)… | ATGATTACAA… |
| *ahpC* | - | 1.43e-5 | TTCCTCCT (-15 ~ -8 nt)… | ATGTCTTTAA… |
| *serp0483* | + | 2.12e-5 | TACCTCCT (-134 ~ -127 nt)… | ATGAAAAATA… |
| *srrAB* | - | 2.12e-5 | TACCTCCT (-17 ~ -10 nt)… | ATGACTAACG… |
| *qoxB* | - | 2.81e-5 | GACCTCCT (-15 ~ -8 nt)… | GTGTCAAAAT… |
| *serp1797* | - | 5.10e-5 | CACCTCAT (-15 ~ -8 nt)… | ATGAAAAAAC… |
| *ctaA* | - | 5.10e-5 | CACCCCAT (-15 ~ -8 nt)… | TTGTTTAGAA… |
| *atlE* | - | 5.26e-5 | CTCCTCGC (-18 ~ -11 nt)… | ATGGCGAAAA… |
| *icaA* | - | 9.90e-5 | CACCTACC (-13 ~ -6 nt)… | ATGCATGTAT… |

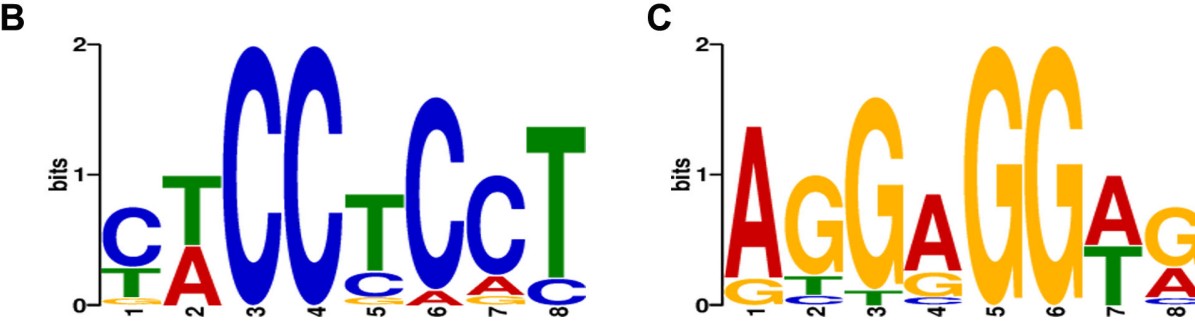

**FIG 10** Schematic diagram of the SrrA promoter sequence in *S. epidermidis*. The suspected target genes directly regulated by SrrA were gathered, and a motif-based sequence analysis was performed at https://meme-suite.org/meme/. The SrrA binding motif (or box) was present in the proximal promoter region and almost located −4 bp to −18 bp upstream of the target genes at the transcription start site. The SrrA box with a high likelihood (*p*-value) was 8 nt in width (A). The SrrA binding box was shown as the standard pattern (B) or the reverse complement pattern (C). The conservative property of each base was indicated by the heights of each letter.

*srrAB* impaired the ability to scavenge ROS and led to the accumulation of ROS in the staphylococcal growth. The ROS levels during the different growth phases were further detected. They showed that the ROS accumulation in the cultures of the Δ*srrAB* and Δ*srrA* mutants was increased significantly compared to that of the parent strain SE1457, especially in the post-stationary phase. The ROS level of the *srrAB* deletion mutant under the oxic condition was higher than that under the microaerobic condition (Fig. 3). These results may contribute to the decreased resistance of the Δ*srrAB* mutant against ROS and indicated that SrrAB was involved in the modulation of endogenous oxidative stress in *S. epidermidis*.

Additionally, the growth curves from different initial bacterial inoculums showed the similarities, such as the *srrAB* deletion mutant being constantly delayed by about 3–4 h compared to the parent strain SE1457. This time lag was partly (not completely) restored by the shift from oxic to microaerobic conditions, which indicated that the reason for the growth retardation between the *srrAB* deletion mutant and the parent strain SE1457 was the result of the combination of two factors. Before entering the log phase, the low expression of the metabolic genes mentioned previously in the *srrAB* deletion mutant

may be the main reason. After entering the log phase, the decreased resistance against ROS may play a more important role in the growth of the Δ*srrAB* mutant due to an increase in ROS accumulation. Notably, in the post-stationary phase, the optical density of the *srrAB* deletion mutant was decreased more rapidly (Fig. 2, 3, and 5).

Once engulfed by phagocytes, bacteria must overcome the exogenous oxidative stress generated due to respiratory bursts. Although there was no significant difference in the phagocytosis ratio between the SE1457 *srrAB* isogenic mutants, the deletion of *srrAB* led to decreased intracellular survival of *S. epidermidis* in the mouse macrophage Ana-1 (Fig. 6). It was speculated that the decreased survival of the Δ*srrAB* mutant compared with that of the parent strain SE1457 may be due to the intracellular accumulation of ROS and decreased ability to scavenge ROS. This speculation was further confirmed by DCFH-DA staining that the ROS-positive cells infected by the *srrAB* deletion mutant had about a twofold increase compared to those infected by its parent strain and complementation strain using a flow cytometer, which was consistent with results detected using a fluorescence microplate reader (Fig. 7). Thus, *S. epidermidis* SrrAB was also involved in modulation of exogenous oxidative stress.

Beyond these common enzymes KatA, AhpC, and ScdA, the thioredoxins were major contributors to oxidative stress resistance by facilitating the reduction of $H_2O_2$, scavenging $HO^-$, and donating reducing equivalents to peroxiredoxins and peroxidase (41, 42). With or without oxidative stressor treatment, the expressions of the ROS-scavenging genes, such as *katA*, *ahpC*, *scdA*, *serp1797* (NAD-dependent protein deacetylase), and *serp0483* (thioredoxin, putative), were downregulated in the Δ*srrAB* mutant (Fig. 8; Table 1), which was attributed to its reduced ability to detoxify oxidative stress, then exhibited decreased resistance against oxidants ($H_2O_2$, CHP) and killing of macrophages. However, the transcription levels of *sodA*, *serp1478* (Dps homolog), *serp1398* (PerR homolog), *abfR* (MgrA homolog), and SarA (transcriptional regulator) in the *srrAB* deletion mutant were not considerably changed. It indicated that these genes were not directly under the control of the SrrAB regulon in *S. epidermidis*, which differed from that of *S. aureus* (26, 28). The interaction between SrrA and its putative promoter region was further explored (Fig. 9). An EMSA showed that phosphorylated SrrA bound to the promoter regions of *katA*, *ahpC*, *scdA*, *serp1797*, and *serp0483*, respectively, which was different from the study reported by Mashruwala et al. (26) that SrrA only bound to the promoter region of *dps* in *S. aureus*. A previous study (27) also found that *S. epidermidis* SrrA is also bound to the promoter regions of genes related to growth and biofilm formation. Combined with these results, we found an 8 bp conserved motif (referred to as SrrA box, YWCCTCCT) in 13 of the 13 putative regions of the SrrAB regulon (Fig. 10). The extended pattern provided more insights into SrrAB regulation in *S. epidermidis*.

In addition to the ROS scavenging genes that were differentially expressed, the genes involved in maintaining metal ion homeostasis or a reduced state in the Δ*srrAB* mutant were significantly changed, such as *serp1120* (transcriptional regulator, Fur family), *cysIJ* (sulfite reductase flavoprotein), *serp1978* (nitroreductase family protein), *isdG* (heme-degrading monooxygenase), *serp1039* (menaquinone biosynthesis methyltransferase), *serp1777* (ferric citrate ABC transporter), *mnhBC* (monovalent cation/proton antiporter), and *sitB* (iron-Chelated ABC transporter) (Table 1). The imbalance in ion transport and the reducing environment may influence the enzymes and Fenton chemistry activities, followed by a high ROS burden on the *srrAB* deletion mutant. Therefore, further studies are necessary.

In summary, SrrAB influences the resistance and intracellular survival of *S. epidermidis* against oxidative stress by regulating the transcription levels of the genes involved in ROS scavenging and ion homeostasis, by which *S. epidermidis* detoxifies and adapts to the commensal environment full of ROS.

## MATERIALS AND METHODS

### Bacterial strains, plasmids, and cell lines

The strains and plasmids used in this study are presented in Table 2 (27, 43–49). Yi Cun Gao from Hong Kong University kindly provided *S. epidermidis* 1457 (SE1457) and *S. aureus* RN4220 (50). The staphylococcus used in the experiment was cultured using TSB (OXID) or TSA (Solarbio) at 37°C. For aerobic incubation, *S. epidermidis* strains were cultured overnight in 15 mL culture tubes with a culture volume of 2 mL. To maintain the plasmid in *S. epidermidis*, researchers supplemented the medium with 5 µg/mL erythromycin. *E. coli* strains were routinely cultivated in Luria-Bertani (LB) medium (10 g tryptone, 5 g yeast extract, and 10 g NaCl) supplemented with 10 µg/mL chloramphenicol or 50 µg/mL kanamycin where appropriate. The mouse macrophage Ana-1 cells were cultured at 37°C with 5% $CO_2$ in RPMI-1640 medium supplemented with 10% fetal bovine serum (VivaCell, Shanghai, China), 100 U/mL penicillin, and 50 µg/mL streptomycin.

### Extraction of bacterial DNA

The genomic DNA of *S. epidermidis* was extracted as described by Flamm et al. (51) with minor modifications. In brief, staphylococcal cells were treated with lysostaphin (20 µg/mL, Sigma) and proteinase K (100 µg/mL) and extracted using the FastPure Bacteria DNA Isolation Mini Kit (Vazyme, Nanjing, China) according to the manufacturer's instructions. According to the manufacturer's instructions, plasmid DNA from *E. coli* was extracted with an EndoFree Maxi Plasmid Kit (TIANGEN, Beijing, China). In particular, plasmid DNA from *S. epidermidis* or *S. aureus* RN4220 was extracted using the same method except for an additional step of lysostaphin and proteinase K treatment.

### Construction of *S. epidermidis* *srrAB* deletion mutant and complementary strains

The *srrAB* deletion mutant of SE1457 was constructed by homologous recombination using the temperature-sensitive plasmid pKOR1 as described by Bae with minor modification (52). In brief, the 988 bp downstream fragment of *srrAB* was PCR-amplified from SE1457 genomic DNA using primer pair srrA-DS-F/srrA-DS-R, and the 951 bp upstream fragment of *srrAB* was amplified using primer pair srrA-US-F/srrA-US-R (sequences listed in Table 3). PCR products were ligated after digestion with *KpnI*, then

**TABLE 2** Bacterial strains and plasmids used in this study[a]

| Plasmid or strain | Description | Source or reference(s) |
|---|---|---|
| **Bacterial strains** | | |
| *S. epidermidis* RP62A | Standard strain of *S. epidermidis*, biofilm positive | (43, 44) |
| *S. epidermidis* 1457 | Biofilm positive, clinical isolate, and wild-type strain | (45, 46) |
| Δ*srrA* mutant | *srrA* deletion, Spc[r], derivative of *S. epidermidis* 1457 | (27) |
| Δ*srrAB* mutant | *srrAB* deletion, derivative of *S. epidermidis* 1457 | This study |
| Δ*srrAB*(pCN51-*srrAB*) mutant | Δ*srrAB* strain complemented with plasmid pCN51-*srrAB* | This study |
| Δ*srrAB*(pCN51) mutant | Δ*srrAB* mutation introduced with plasmid pCN51 | This study |
| *S. aureus* 4220 | Restriction negative and modification positive | (47, 48) |
| *E. coli* DH5α | supE44 ΔlacU169 (Φ80dlacZΔM15) hsdR17 recA1 endA1 gyrA96 thi-1 relA1 | Invitrogen |
| *E. coli* BL21(DE3) | F⁻ ompT hsdSB(rB⁻mB⁻) gal dcm (DE3) | Invitrogen |
| **Plasmids** | | |
| pKOR1 | Temp-sensitive *E. coli* (Amp[r])-*Staphylococcus* (Cm[r]) shuttle vector | Li Ming Fudan University |
| pKOR1-ΔsrrAB | Recombinant plasmid | This study |
| pET28a | *E. coli* expression plasmid; Km[r] | Novagen |
| pET28a-srrA | pET28a harboring the srrA gene, used for SrrA expression | (27) |
| pCN51 | Shuttle vector; Amp[r] Em[r] | (49) |
| pCN51-srrAB | The srrAB gene was cloned into pCN51 | This study |

[a]Amp[r], ampicillin resistance; Cm[r], chloramphenicol resistance; Em[r], erythromycin resistance; Km[r], kanamycin resistance; Spc[r], spectinomycin resistance.

cloned into a pKOR1 vector with BP clonase enzyme (Invitrogen) to yield a replacement plasmid pKOR1-ΔsrrAB. The recombinant plasmid pKOR1-ΔsrrAB was successfully transferred into *E. coli* DH5α, *S. aureus* RN4220, and then SE1457. The allelic replacement was performed as described previously. The *srrAB* deletion mutant (ΔsrrAB) was verified by PCR, RT-PCR, and sequencing.

For complementation of the ΔsrrAB mutant, the *srrAB* gene with the associated Shine-Dalgarno sequence in SE1457 was amplified by PCR with primers as pCN51-*srr*-F/pCN51-*srr*-R. The pCN51-*srrAB* was constructed from pCN51 inserted with a fragment of *srrAB* digested with *KpnI* and *BamHI*. The complementary plasmid was transferred into ΔsrrAB by electroporation, yielding complementary strain ΔsrrAB(pCN51-*srrAB*). The vector plasmid pCN51 was introduced as blank control into ΔsrrAB, named ΔsrrAB(pCN51). In addition, the *srrA* deletion mutant (ΔsrrA) derived from SE1457 was constructed by allelic replacement using the temperature-sensitive plasmid pMAD as described previously (27, 53), and the ΔsrrA mutant as control was carried out in parallel for all experiments.

## Bacterial viability assay

The bacterial viability in the culture was evaluated by the colony counting method. In brief, the SE1457 *srrAB* isogenic mutants were diluted (1:200) in fresh TSB medium and incubated at 37°C with shaking. Overnight cultures were removed from the incubator and kept at room temperature for observation. At each point (0, 6, 12, 24, 48, 72 h), 0.5 mL of bacterial suspension was centrifuged, washed twice with normal saline, and serially diluted (10-fold). Each aliquot of 100 µL was spotted onto a TSA plate for CFU counting (three petri dishes per dilution). Four independent experiments were carried out.

## Plate assay and growth curves for oxidative sensitivity

For the stress plate assay, overnight cultures of *S. epidermidis* strains were diluted 1:200 with fresh TSB medium and incubated at 37°C with aeration for 4 h until $OD_{600}$ was approximately 0.8. After several 1:10 serial dilutions, each aliquot of 5 µL was spotted onto a TSA plate in the presence of either $H_2O_2$ or CHP at varying concentrations (0.25–1.0 mM). The cell-permeable metal chelator DIP was added to a final 1–2 mM concentration in the TSA plate when appropriate. For static incubation under microaerobic conditions, the plates inoculated with bacteria were placed in an anaerobic culture tank (MART, Holland) filled with a mixed gas of 85% $N_2$, 5% $O_2$, and 10% $CO_2$. Staphylococcal growth on the oxidant-containing medium was recorded after incubation at 37°C for 24 h. In a set of assays, the TSA plate without oxidative stress was set and used in the $H_2O_2$ and CHP stress assay.

According to the manufacturer's instruction, the growth curves of *S. epidermidis* strains challenged with or without $H_2O_2$ were determined in a SpectroSTAR Nano Plate Reader (BMG LabTech, Ortengerg, Germany). Overnight cultures were diluted (1:200) into fresh TSB medium and incubated at 37°C with aeration for 4 h until $OD_{600}$ was approximately 0.8. After 10-fold serial dilutions, the bacterial suspension was inoculated (1:200) into a TSB medium with or without 1 mM $H_2O_2$. For oxic conditions, the bacterial suspension was added to triplicate wells (200 µL/well) in a 96-well plate and placed into a heated microplate reader that allowed for free diffusion of gases. For microaerobic conditions, the bacteria were cultured into a 96-well plate completely filled with the medium, and the plate was sealed with sealing film (Axygen, Union City, CA, United States). Finally, the plates were incubated at 37°C with shaking at 200 rpm. The $OD_{600}$ values of the cultures were measured at 60 min intervals for 12 h or 24 h with the plate reader.

## Detection of reactive oxygen species

The ROS levels during bacterial growth were determined using NBT reduction as described by Hussain et al. (54) with minor modification. In brief, overnight cultures

**TABLE 3** Primers used in this study[a]

| Method and primer | Sequence (5'–3')[b] | Location (bp)[c] | Restriction enzyme | Product size (bp) |
|---|---|---|---|---|
| Construction and identification of the *srrAB* deletion mutant | | | | |
| srrAB-DS-F | GGGGACAAGTTTGTACAAAAAAGCAGGCTATAATGG AAGTAACACAAAATAATT | 1098982–1099006 | attB | 988 |
| srrAB-DS-R | GGGGTACCAGTTAGAAACTGAAAAGTATCATA | 1099946–1099969 | kpnI | 988 |
| srrAB-US-F | GGGGTACCagtcatactttctactacct | 1102434–1102453 | kpnI | 951 |
| srrAB-US-R | GGGGACCACTTTGTACAAGAAAGCTGGGTGCATTAC CTACTACAGAAG | 1103371–1103389 | attB | 951 |
| *srrAB* complementation | | | | |
| pCN51-srrAB-F | CGCGGATCCACTCAATAACGTTAACCTATGATAT | 1102485–1102509 | BamHI | 2,563 |
| pCN51-srrAB-R | CGGGGTACCATGATACTTTTCAGTTTCTAACTAA | 1099947–1099971 | KpnI | 2,563 |
| Transcriptional analysis by qRT-PCR | | | | |
| srrA-F | TCACCTAGAGAAGTAGTATT | 1102108–1102127 | | 129 |
| srrA-R | GAGCGTCATTATCAATCA | 1101998–1102015 | | 129 |
| srrB-F | TCCATAGTAGACGGTATAGT | 1100559–1100578 | | 135 |
| srrB-R | ATAATCCTTCAGCATCCATA | 1100443–1100462 | | 135 |
| katA-F | AACTATACTGACGAGGAAG | 915225–915243 | | 159 |
| katA-R | AAGGATTGTCTGGATGATT | 915366–915384 | | 159 |
| ahpC-F | AACTTCACCTGGATGTTG | 47937–47954 | | 90 |
| ahpC-R | AATCAATGCTGACGGAAT | 48010–48027 | | 90 |
| scdA-F | CCTTGACTATATTGAATGAG | 329038–329057 | | 165 |
| scdA-R | AGAATCTTACACCTTACAT | 329185–329203 | | 165 |
| SERP0483-F | CAACTTGTTCAGGTGATT | 478434–478451 | | 191 |
| SERP0483-R | CGATGGATATGTGGATAGA | 478607–478625 | | 191 |
| SERP1797-F | CAGTCTTCAAGGCATTAG | 1843309–1843326 | | 185 |
| SERP1797-R | TAGTCATATCATCGTGTATAAC | 1843473–1843494 | | 185 |
| SERP0760-F | GCCTACATTGACTACATT | 759333–759350 | | 185 |
| SERP0760-R | ATTCTAATTCTGCCTTCTT | 759500–759518 | | 185 |
| nrdD-F | CCATATTGACTGCTTGAA | 2217135–2217152 | | 175 |
| nrdD-R | GACTTAGATTACCATCCATT | 2217291–2217310 | | 175 |
| nrdG-F | ACCATCAACTAATACATCAA | 2215248–2215267 | | 151 |
| nrdG-R | TTATCACACTCCAACTTG | 2215382–2215399 | | 151 |
| pflA-F | CTTCTCTTGATGGTTCGTTA | 2411983–2412002 | | 139 |
| pflA-R | ACACTTACACTCCGTTGA | 2412105–2412122 | | 139 |
| pflB-F | AATACCTACACCACCAATAG | 2413388–2413407 | | 147 |
| pflB-R | TGACATCACTGAACAAGAA | 2413517–2413535 | | 147 |
| SERP0815-F | GCCATTGTTATCATTATCCT | 819261–819280 | | 176 |
| SERP0815-R | TTCTTCAGCCTCAGTTAT | 819419–819436 | | 176 |
| SERP1273-F | TTATTGTTGGCTCTGTATCA | 1318416–1318435 | | 108 |
| SERP1273-R | GTGTTGCTACTTGAGGTT | 1318506–1318523 | | 108 |
| sodA-F | CCACCGCCATTATTACGA | 1157642–1158659 | | 1,106 |
| sodA-R | GCAGTTGAAGGGACAGAT | 1158731–1158748 | | 1,106 |
| codY-F | CTATAACAATGGCAATCAACTC | 826859–826880 | | 181 |
| codY-R | TCTATGACACCAGCACTT | 827022–827039 | | 181 |
| fhs-F | GCAACAGTAACATTATGAT | 1342739–1342757 | | 138 |
| fhs-R | GATACAAGAACAGGAGAA | 1342859–1342876 | | 138 |
| Amplification of promoter fragments | | | | |
| B$_{srrA}$-F | GGCCCTATAGATTTAAAAG | 1102653–1102671 | | 269 |
| B$_{srrA}$-R | CTATCTTCATCATCAACGA | 1102402–1102420 | | 269 |
| B$_{katA}$-F | GCCGTCTCCCATCTATCT | 914325–914342 | | 248 |
| B$_{katA}$-R | TTTTCCATCCTGTTTTGAC | 914555–914573 | | 248 |
| B$_{ahpC}$-F | ATCAATAAAATAACCATAG | 48717–48735 | | 303 |
| B$_{ahpC}$-R | AGACATAGATAAATTCCTC | 48432–48450 | | 303 |

(*Continued on next page*)

TABLE 3 Primers used in this study[a] (*Continued*)

| Method and primer | Sequence (5′–3′)[b] | Location (bp)[c] | Restriction enzyme | Product size (bp) |
|---|---|---|---|---|
| B$_{scdA}$-F | ATTCTGGGTTAGCCTCAAA | 329754–329772 | | 291 |
| B$_{scdA}$-R | AATCATAAAAATTCCTCCT | 329481–329499 | | 291 |
| B$_{SERP1797}$-F | TTGATTTATTCACTTCCTA | 1842489–1842507 | | 337 |
| B$_{SERP1797}$-R | ACGATATCTTTTAACTGTT | 1842808–1842826 | | 337 |
| B$_{SERP0483}$-F | ATTCACACCCGTTTTATCT | 479005–479023 | | 303 |
| B$_{SERP0483}$-R | TTTCATCACTATCTACTCC | 478720–478738 | | 303 |
| B$_{rpsJ}$-F | GGAAAACCTTGAATTATCA | 1862576–1862594 | | 283 |
| B$_{rpsJ}$-R | GAAACATCTGCACCAGAAC | 1862311–1862329 | | 283 |

[a]The primers were designed using Primer Premier five software according to the genomic sequence of *S. epidermidis* RP62A (GenBank accession number NC_002976).
[b]Restriction sites are indicated by underlining.
[c]The locations of primers are indicated according to the *S. epidermidis* RP62A genome.

were diluted (1:200) into fresh TSB medium and incubated at 37°C with aeration for 4 h (OD$_{600}$ of 0.8). The bacterial suspension was adjusted to $1.0 \times 10^6$ CFU/mL and serially diluted (10-fold), then pipetted into the microplate, and the subsequent procedures were the same as those used for the growth curve assay in a SpectroSTAR Nano Plate Reader. Each time, 100 µL of bacterial suspension was withdrawn, and 0.5 mL NBT (1 mg/mL) was added. After 30 min incubation at 37°C, 100 µL of HCl (0.1 mol/L) was added, followed by centrifugation at 1,500 rpm for 10 min, and then the ROS levels in the colored supernatants were measured with OD$_{575}$ value.

The ROS levels in the macrophage were determined using DCFH-DA (Solarbio, Beijing, China) as previously described by Dwivedi et al. (55). In brief, mouse macrophage Ana-1 cells were seeded in a six-well plate at a density of $1.5 \times 10^7$ cells/mL, then 1 mL of the cell suspension was incubated with the staphylococci (1 mL, $1.5 \times 10^8$ CFU/mL) at a multiplicity of infection (MOI) of 10 in a six-well plate at 37°C for 6 h with 5% CO$_2$. The Ana-1 cells were collected by centrifugation at 800 rpm for 3 min and washed twice with PBS, then treated with 100 µg/mL gentamycin and 20 µg/mL lysostaphin for 30 min at 37°C with 5% CO$_2$. The infected Ana-1 cells were washed twice with PBS again and resuspended in serum-free RPMI-1640 medium (1.5 mL) containing 10 µmol/L DCFH-DA, then incubated at 37°C in the dark for 20 min with 5% CO$_2$. After incubation, the infected Ana-1 cells were washed (800 rpm, 3 min) three times with serum-free RPMI-1640 medium to remove any remaining DCFH-DA. Two milliliter of the cell suspension was added into a six-well plate and observed under an inverted fluorescence microscope AxioCam 705 mono (Zeiss, Oberkochen, Germany). The fluorescence intensity, correlated with the intracellular ROS level, was measured using a FACSCanto II flow cytometer (BD Pharmingen, Heidelberg, Germany) with the same measurement parameters as fluorescein isothiocyanate (FITC) and using a Multi-mode Microplate Reader (Synergy HT, Bio-Tek, USA) at an excitation wavelength of 488 nm and emission wavelength of 525 nm.

## Phagocytosis and bactericidal assay by macrophages

As described previously, *S. epidermidis* phagocytosis assays were performed with minor modifications (48). In brief, staphylococci in the exponential growth phase were adjusted to a 0.5 McFarland standard ($1.5 \times 10^8$ CFU/mL) with RPMI-1640 medium, and CFU counting was performed at an initial incubation time point (CFU$_{t0}$). Mouse macrophage Ana-1 cells were washed twice with RPMI-1640 medium without antibiotics and adjusted to $1.5 \times 10^7$ cells/mL, and then 1 mL of the cell suspension was co-incubated with the staphylococci (1 mL) at a MOI of 10 in a six-well plate at 37°C with 5% CO$_2$. After 6 h incubation, the culture was centrifuged at 800 rpm for 3 min, and the bacterial suspensions were removed. The cells were collected and serially diluted to 10-fold, and then the unengulfed bacteria were determined by CFU counting (CFU$_{t1}$). The Ana-1 cells were washed twice ($800 \times g$, 3 min) with normal saline, then treated with 100 µg/mL gentamycin and 20 µg/mL lysostaphin for 30 min to kill the extracellular bacteria. The

Ana-1 cells were lysed with RIPA lysis buffer (Solarbio, Beijing, China), and the number of surviving staphylococci was evaluated by CFU counting ($CFU_{t2}$). The phagocytosis rate was evaluated by the equation of ($CFU_{t0} - CFU_{t1}$)/$CFU_{t0}$.

## RNA isolation and RNA sequencing

Total RNA was extracted by using the RNeasy Mini kit (QIAGEN, Hilden, Germany) according to the manufacturer's instructions. In brief, the overnight cultures of *S. epidermidis* and the derivative strains were diluted 1:200 into 20 mL TSB medium and incubated at 37℃ with shaking. These log-phase cells ($OD_{600}$ reached 0.8–1.0) were harvested after 6 h incubation and washed twice with ice-cold normal saline. For the *srrAB* transcription level detected under oxidative stress, the SE1457 strain was pre-treated with different concentrations of $H_2O_2$ and CHP for 30 min before harvest. The cell pellets were homogenized five times with 0.5 mL of 0.1 mm Zirconia-silica beads in a Mini-Beadbeater (Biospec, Bartlesville, OK, USA) at a speed of 4,800 rpm for 40 s at 1 min intervals on ice. The samples were centrifuged at 12,000 rpm for 10 min, and then RNA in the supernatant was extracted using the silica-based filter of the RNeasy Mini kit.

RNA-seq analysis was conducted according to the Illumina RNA sequencing sample preparation guide as previously described by Wang et al. (56). In brief, RNA samples were digested with RNase-free DNase I (Sigma, St. Louis, MO, USA) to remove the genomic DNA. The RNA concentration and quality were assessed using a Qubit 3.0 Fluorometer (Life Technologies, USA) and Nanodrop One spectrophotometer (Thermo Fisher Scientific Inc., USA). The integrity of total RNA was assessed using a BioAnalyzer system (Agilent Technologies Inc., USA), and samples with RNA integrity number values above 7.0 were used for sequencing. One microgram of RNA was used as input material for the RNA sample preparations, and rRNA was removed with a Ribo-off rRNA Depletion kit (Bacteria; Vazyme, China). The rRNA was hybridized with DNA probes, and the DNA/RNA hybridization strand was digested with RNase H and DNase I, respectively. Then, RNA was purified by magnetic beads after the removal of rRNA. The obtained RNA was fragmented and PCR amplified with random primers using a Whole RNA-seq Lib Prep Kit (ABclonal, China). The cDNA libraries were prepared in a strand-specific manner with the same kit. Purified libraries were quantified and validated by Qubit 3.0 Fluorometer and Agilent 2100 BioAnalyzer to confirm the insert size and calculate the mole concentration. The cluster was generated by cBot after the library was diluted to 10 pM and then sequenced on the Illumina NovaSeq 6000 platform (Illumina, USA).

After the removal of rRNA reads, sequencing adapters, other lower-quality reads, and the remaining reads were multi-mapped to the reference genome of *S. epidermidis* RP62A at the NCBI website using Bowtie2 software. The differential expression of different transcripts was quantified using DEGseq software. A gene with a threshold of false discovery rate-adjusted *P*-value less than 0.05 (*t*-test) and at least a 2.0- or 0.5-fold change in transcript level was designated as significant differences in expression ratios.

## Quantitative real-time reverse transcription-PCR

The RNA extracted from SE1457 and the Δ*srrAB* mutant was reverse transcribed into cDNA using HiScript III RT SuperMix for quantitative PCR (qPCR; Vazyme, Nanjing, China). Then, qPCRs were performed using ChamQ Universal SYBR qPCR Master Mix (Vazyme, Nanjing, China) in a Stepone Real-Time PCR System (Applied Biosystems, USA). The amplification conditions were 95℃ for 30 s, 40 cycles of 95℃ for 10 s, and 60℃ for 30 s, followed by melting-curve analysis. The reactions were normalized using *gyrB* (DNA gyrase subunit B) as the housekeeping gene. All qRT-PCRs were performed in triplicate and repeated at least three times. The sequences of the primers were designed using Beacon Designer software (Premier Biosoft International, Palo Alto, CA) and are listed in Table 3.

## Electrophoretic mobility shift assay

His-tagged SrrA was purified as previously described by Wu et al. (27), and EMSA was performed as described previously using biotin 5′-end labeled promoter probes with minor modification. In brief, the biotin-labeled DNA probes containing the putative promoter sequences of *srrAB*, *katA*, *ahpC*, *scdA*, *serp1797*, and *serp0483* (248–337 bp fragments) were amplified by PCR from SE1457 genomic DNA with biotin-labeled primers listed in Table 3. The DNA fragments were purified using a Universal DNA Purification Kit (Tiangen, Beijing, China). Purified His-tagged SrrA was phosphorylated (SrrA-P) by incubation with 50 mM acetylphosphate (Sigma, Steinheim, Germany) for 1 h at room temperature. Each gel shift assay included the probe labeled with biotin plus increasing concentrations of SrrA-P (ranging from 50 to 250 µM), and a 125-fold molar excess of the unlabeled specific probe was added into the labeled probe plus 250 µM SrrA-P as a competitor, and a 125-fold molar excess of unlabeled nonspecific DNA (283 bp coding sequence of *rpsJ*) was added into the labeled probe plus 250 µM SrrA-P as a negative control. All samples were incubated at 25°C for 30 min in 20 µL of EMSA/Gel-Shift Binding Buffer (Beyotime Biotech, Shanghai, China). Following incubation, the protein-probe mixtures were separated by electrophoresis on a 5% nondenaturing polyacrylamide gel in 0.5× Tris-borate-EDTA (TBE) buffer and blotted onto a positively charged nylon membrane (Millipore, Bedford, MA, USA). Migration of biotin-labeled probes was detected by horseradish peroxidase-conjugated streptavidin that binds to biotin and chemiluminescent substrate according to the manufacturer's instruction and then imaged using an Image Quant LAS 4000 Mini Biomolecular Imager (GE Healthcare, USA).

## Statistical analysis

Data from the bacterial viability detection, qRT-PCR assay, phagocytosis and bactericidal assay, and detection of reactive oxygen species were analyzed by GraphPad Prism software (San Diego, CA, USA) using the Student's $t$-test or one-way ANOVA. Differences with a $P$ value of less than 0.05 were considered statistically significant.

## ACKNOWLEDGMENTS

This work was supported by the National Natural Science Foundation of China (Nos. 82060380 and 81660346 to Youcong Wu; No. 82472326 and 82072249 to Yang Wu) and the Young and the Middle-aged Academic Leader Training Foundation of Yunnan Province (No. 202305AC160038 to Youcong Wu).

## AUTHOR AFFILIATIONS

[1]Department of Medical Microbiology and Immunology, School of Basic Medical Sciences, Health Science Center, Dali University, Dali, Yunnan, China
[2]Key Laboratory of Medical Molecular Virology (MOE/NHC/CAMS), Shanghai Frontiers Science Center of Pathogenic Microorganisms and Infection, School of Basic Medical Sciences, Shanghai Medical College, Fudan University, Shanghai, China

## AUTHOR ORCIDs

Chunjing Zhao  http://orcid.org/0009-0005-9316-2889
Yang Wu  http://orcid.org/0000-0003-1358-2807
Youcong Wu  http://orcid.org/0000-0002-7809-5850

## AUTHOR CONTRIBUTIONS

Chunjing Zhao, Conceptualization, Data curation, Formal analysis, Investigation, Methodology, Software, Supervision, Validation, Visualization, Writing – original draft, Writing – review and editing | Youcong Wu, Resources, Writing – review and editing, Funding acquisition, Project administration, Supervision.

## DATA AVAILABILITY

The complete RNA-seq data set is posted in the Gene Expression Omnibus database (http://www.ncbi.nlm.nih.gov/geo/) under accession numbers GPL34913 for the platform design and GSE277473 for the original data set.

## ADDITIONAL FILES

The following material is available online.

### Supplemental Material

**Fig. S1 (mSystems01737-24-s0001.tif).** Construction of *srrAB* deletion in SE1457 by allelic replacement.

**Fig. S2 (mSystems01737-24-s0002.tif).** Viability detection by CFU counting.

**Legends (mSystems01737-24-s0003.docx).** Legends for supplemental figures and tables.

**Table S1 (mSystems01737-24-s0004.docx).** Effect of *srrAB* deletion on the intracellular ROS production in *S. epidermidis*-infected Ana-1 cells

**Table S2 (mSystems01737-24-s0005.pdf).** RNA Seq analysis of the *srrAB* deletion mutant in *S. epidermidis*.

### Open Peer Review

**PEER REVIEW HISTORY (review-history.pdf).** An accounting of the reviewer comments and feedback.

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
