## [Reviewer comments · mSystems]

Staphylococcus epidermidis uses the SrrAB regulatory system to modulate oxidative stress and intracellular survival in mouse macrophage cell line Ana-1

Chunjing Zhao, Zongkai Bai, Xiaoting Chen, Shuangjie Shang, Baitong Shen, Li Bai, Di Qu, Yang Wu, and Youcong Wu

Corresponding Author(s): Chunjing Zhao, Dali University School of Basic Medicine

Review Timeline:

Submission Date:	December 19, 2024
Editorial Decision:	February 14, 2025
Revision Received:	February 17, 2025
Accepted:	February 24, 2025

Editor: Ying Zhang

Reviewer(s): The reviewers have opted to remain anonymous.

Transaction Report:

DOI: <https://doi.org/10.1128/msystems.01737-24>

Re: mSystems01737-24 (Staphylococcus epidermidis uses the SrrAB regulatory system to modulate oxidative stress and intracellular survival in mouse macrophage cell line Ana-1)

Dear Ms. Chunjing Zhao:

Please address the reviewer's questions and correct any remaining grammatical errors (e.g. Ln 74, 75, etc).

Revision Guidelines

Sincerely,
Ying Zhang
Editor
mSystems

Reviewer #2 (Comments for the Author):

This manuscript has been reviewed three times and I appreciate the many significant changes the authors have made to clarify their findings and demonstrate the rigor of their approaches. However, there are two minor points from the last review that remain unaddressed in the text of the manuscript (below). While many grammatical errors have been addressed, a number still remain.

Line 75: "AbfR is the first oxidation sensor of *S. epidermidis*"

Do the authors mean that this is the first described sensor of oxidative stress or that it has been demonstrated that AbfR must act first to respond to oxidative stress? Please clarify.

Line 563-4: "The cDNA libraries were prepared by using an Whole RNA-seq Lib Prep Kit (ABclonal, China)." Can you please clarify if these libraries were made in a strand-specific manner?

Dear Prof. Zhang

We are truly grateful to yours and other reviewer's critical comments and thoughtful suggestions. They have been invaluable in helping us improving our work and gain a deeper understanding of the areas that require refinement. Your insights have been a guiding light throughout the revision process, and we sincerely appreciate the time and effort you have invested in providing such constructive feedback. All changes made to the text are in red color. We hope the new manuscript will meet the journal's standard. Below you will find out point-by-point responses to the reviewer's comments or questions:

Reviewer #2 (Comments for the Author):

This manuscript has been reviewed three times and I appreciate the many significant changes the authors have made to clarify their findings and demonstrate the rigor of their approaches. However, there are two minor points from the last review that remain unaddressed in the text of the manuscript (below). While many grammatical errors have been addressed, a number still remain.

Reply: We are very sorry for the grammatical mistakes in the manuscript. We have checked the grammar and expression of the whole text. The modified parts have been marked by red in the revised manuscript. Thank you again for your attention to our manuscript.

Line 75: "AbfR is the first oxidation sensor of *S. epidermidis*"

Do the authors mean that this is the first described sensor of oxidative stress or that it has been demonstrated that AbfR must act first to respond to oxidative stress? Please clarify.

Reply: We are very sorry for the confused statement. AbfR has been firstly identified as an oxidation-sensing regulator that regulates bacterial aggregation and biofilm formation by responding to oxidative stress in *S. epidermidis* according to the Liu's finding (The reference is below). This sentence "AbfR is the first oxidation sensor of *S. epidermidis* and regulates oxidative stress responses, bacterial aggregation, and biofilm formation" was changed to "AbfR is the first described sensor of oxidative stress in *S. epidermidis* and regulates oxidative stress responses, bacterial aggregation, and biofilm formation" in Line 77, and shown in red in the text.

Reference: Xing Liu, Xiaoxu Sun, Youcong Wu, et al. Oxidation-sensing Regulator AbfR Regulates Oxidative Stress Responses, Bacterial Aggregation, and Biofilm Formation in *Staphylococcus epidermidis*[J]. Journal of Biological Chemistry, 2013, 288 (6): 3739-3752.

Line 563-4: "The cDNA libraries were prepared by using an Whole RNA-seq Lib Prep Kit (ABclonal, China)." Can you please clarify if these libraries were made in a strand-specific manner?

Reply: Thank you for your professional and rigorous consideration. The cDNA libraries were constructed in a strand-specific manner. In particularly, the second-strand cDNA was synthesized using the Second Strand Synthesis Reaction

Buffer with dUTP instead of dTTP. Before PCR amplification, the UDG (Uracil-DNA Glycosylase) enzyme can digest the second-strand cDNA, thus ensuring the directionality of the mRNA strand. We have rephrased the sentence as below: “The obtained RNA was fragmented and PCR-amplified with random primers using a Whole RNA-seq Lib Prep Kit (ABclonal, China). The cDNA libraries were prepared in a strand-specific manner with the same kit.” in Line 577-580, and shown in red in the text.

Re: mSystems01737-24R1 (Staphylococcus epidermidis uses the SrrAB regulatory system to modulate oxidative stress and intracellular survival in mouse macrophage cell line Ana-1)

Dear Ms. Chunjing Zhao:

Thank you for addressing the reviewer questions and clear the grammar errors.

Your manuscript has been accepted, and I am forwarding it to the ASM production staff for publication. Your paper will first be checked to make sure all elements meet the technical requirements. ASM staff will contact you if anything needs to be revised before copyediting and production can begin. Otherwise, you will be notified when your proofs are ready to be viewed.

Sincerely,
Ying Zhang
Editor
mSystems